# Division of labor and collective functionality in *Escherichia coli* under acid stress

Sophie Brameyer [1], Kilian Schumacher [1], Sonja Kuppermann[1] & Kirsten Jung [1✉]

The acid stress response is an important factor influencing the transmission of intestinal microbes such as the enterobacterium *Escherichia coli*. *E. coli* activates three inducible acid resistance systems - the glutamate decarboxylase, arginine decarboxylase, and lysine decarboxylase systems to counteract acid stress. Each system relies on the activity of a proton-consuming reaction catalyzed by a specific amino acid decarboxylase and a corresponding antiporter. Activation of these three systems is tightly regulated by a sophisticated interplay of membrane-integrated and soluble regulators. Using a fluorescent triple reporter strain, we quantitatively illuminated the cellular individuality during activation of each of the three acid resistance (AR) systems under consecutively increasing acid stress. Our studies highlight the advantages of *E. coli* in possessing three AR systems that enable division of labor in the population, which ensures survival over a wide range of low pH values.

[1] Department of Biology, Microbiology, Ludwig-Maximilians-Universität München, Martinsried, Germany. ✉email: jung@lmu.de

The acquisition of acid resistance (AR) is an important property of *Escherichia coli* and many other neutralophilic bacteria that enables survival in acidic environments, such as the human gastrointestinal tract or acidic soils[1,2]. Although the cytoplasmic membrane is impermeable for protons, some protons enter the cytoplasm through protein channels, transient water chains, or damaged membranes[3,4]. Bacteria are generally able to maintain a fairly constant internal pH when grown in a wide range of media at different external pH values[5,6]. Despite this, pH homeostasis varies among individual bacterial cells as reported amongst others for *E. coli* and *Bacillus subtilis*[7]. Interestingly, persister cells of *E. coli* display a lower intracellular pH allowing survival after antibiotic treatment[8]. Nevertheless, after exposure of *E. coli* to an external pH of 5.8, the pH of its cytoplasm drops transiently; however, the pH rapidly returns to a neutral level due to the intrinsic buffering capacity of the cytoplasm or alterations in the flux of ions[9]. A lower external pH such as 4.4 causing a lower intracellular pH of around 6.0 has adverse effects on all macromolecules of a cell, which might result in lowered enzyme activity, acid-induced protein unfolding, membrane damage, and DNA damage.

Besides its passive mechanisms, *E. coli* possesses several inducible AR systems to counteract acidic environments[1,3]. The major systems are the glutamate decarboxylase (Gad) system, arginine decarboxylase (Adi) system, and lysine decarboxylase (Cad) system (AR2, AR3, and AR4, respectively)[10–12]. AR1 does not require an amino acid and is regulated by the alternative σ-factor (RpoS) and the cAMP receptor protein[10,13]. The ornithine decarboxylase system (AR5) plays only a minor role in *E. coli* MG1655, yet has a more important role in avian pathogenic *E. coli*[14]. The core components of each of the three major AR systems are a proton-consuming amino acid decarboxylase and a cognate antiporter to excrete the decarboxylated (more alkaline) compounds in exchange with the corresponding extracellular amino acid. In this way, both the intracellular and extracellular pH values can be increased[3,11,15,16]. Each system is active at different external pH values and growth phases, and induction is regulated by a sophisticated interplay of membrane-integrated and soluble regulators.

Activation of the Gad System occurs during the transition of an *E. coli* culture to stationary phase and during exponential growth in acidified media. Furthermore, this system is essential for cell survival at an extremely low pH of 2.5[10,17–19]. The Gad system employs GadA and GadB, two pyridoxalphosphate-dependent decarboxylases that catalyze the proton-consuming decarboxylation of L-glutamate to generate γ-aminobutyrate (GABA), and GadC, the cognate antiporter that performs the import of L-glutamate and the export of GABA[17,20]. It is important to note that the activity of GadC is pH-dependent, and the antiporter preferentially exchanges protonated glutamate ($Glu^0$) in exchange for protonated GABA ($GABA^+$) at an external pH of 3.0 and lower[21–23]. The induction of the *gad* genes is rather complex. The main transcriptional regulator is GadE, whose expression is regulated by the transcription factors EvgA, GadW, GadX, and YdeO as well as σ-factor RpoS. RpoS is activated and stabilized in response to different conditions, including the stationary phase, various stress treatments, and for example by the signal transduction cascade proceeding from PhoQ/PhoP to RssB and IraM[24,25]. Furthermore, the EvgS/EvgA histidine kinase/response regulator system is a primary detector of mild acidic environments (pH 5) and additionally regulates the activation of the Gad system through a cascade of EvgA-YdeO-GadE regulators[26,27] (Fig. 1). In addition, RcsB is a critical partner of GadE and the binding of both regulators as a heterodimer to the GAD box activates *gadA* transcription[28–30].

The Adi system is responsible for the conversion of arginine to agmatine under the consumption of one proton (Fig. 1). The main components are the transcriptional activator AdiY, a member of the AraC-family; the arginine decarboxylase AdiA; and the arginine/agmatine antiporter AdiC. The genes encoding the Adi system have an unusual genomic arrangement in *E. coli*, as the gene of *adiY* is located downstream of *adiA* and upstream of *adiC*. The Adi system is maximally induced under the conditions of acidic pH (pH 4.4), anaerobiosis, and a rich medium[31]. The activity of AdiC is regulated in response to acidic pH and remains fully active at a pH of ≤6.0[32]. The expression of *adiY* seems to be indirectly influenced by the transcriptional repressor CsiR, as its overexpression results in the repression of *adiY* and *adiA*[16]. Under anaerobic conditions, the *adiY* mRNA can be base-paired by the small RNA SgrS resulting in post-transcriptional downregulation[33].

The Cad system, which is activated when *E. coli* is exposed to pH 5.8 in the presence of lysine, uses external lysine, which is converted to cadaverine by the lysine decarboxylase CadA under the consumption of one proton. The lysine/cadaverine antiporter CadB imports lysine and excretes the more alkaline cadaverine, thereby elevating the external pH[34]. Expression of the *cadBA* operon is activated by the membrane-integrated one-component regulator CadC, which is a representative of the ToxR-family[35] (Fig. 1). The pH-sensory function and the feedback inhibition by cadaverine could be assigned to distinct amino acids within the periplasmic sensory domain of CadC[36,37]. The availability of external lysine is transduced to CadC via the co-sensor and inhibitor LysP, which is a lysine-specific transporter[38,39]. Recently, we demonstrated that not only the copy number of CadC affects the dual-stress response[40], but also the noise of the target protein abundance. Using fluorophore-tagged CadB, a heterogeneous output in the single cells of *E. coli* occurred under mild acidic stress. Moreover, an increase in the CadC copy number was correlated with decreased heterogeneous behavior[41].

The ON/OFF behavior of the *E. coli* population in activating the Cad system prompted us to ask whether the "Cad-OFF" cells activate the Gad and/or Adi systems. Here, we used a fluorescent triple reporter to study the induction of the three inducible AR systems, Gad, Adi, and Cad, in individual *E. coli* cells under consecutively increasing acid stress. We herein report extensive heterogeneity and division of labor in the acid stress response of individual *E. coli* cells. For the first time, we present a model that explains the cellular individuality that occurs during activation of each of the three AR systems in the context of ensuring AR for the whole population over a wide range of acidic pH values.

## Results

**Heterogenous activation of the three inducible AR systems by acid stress in *E. coli*.** In our previous studies, we demonstrated that heterogenous activation of the Cad system results in ~70% ON and ~30% OFF cells[41]. Here, we constructed a three-color reporter strain *E. coli* (*gadC:eGFP-adiC:mCerulean-cadB:mCherry*) to study whether the "Cad-OFF" cells activate the Gad and/or the Adi systems and whether these two systems are activated heterogeneously. In this strain, each of the antiporter genes (*gadC*, *adiC*, and *cadB*) is fused with a different fluorophore gene coding for eGFP, mCerulean, or mCherry, respectively (Fig. 1). These fusions were chromosomally integrated to avoid a copy number effect, and the functionality of the hybrid proteins was confirmed (Supplementary Fig. 1).

First, the three-color reporter strain was exposed to consecutively increasing acidic conditions (pH 7.6, pH 5.8, and pH 4.4) in a well-mixed environment (Fig. 2a). Then, at the indicated time

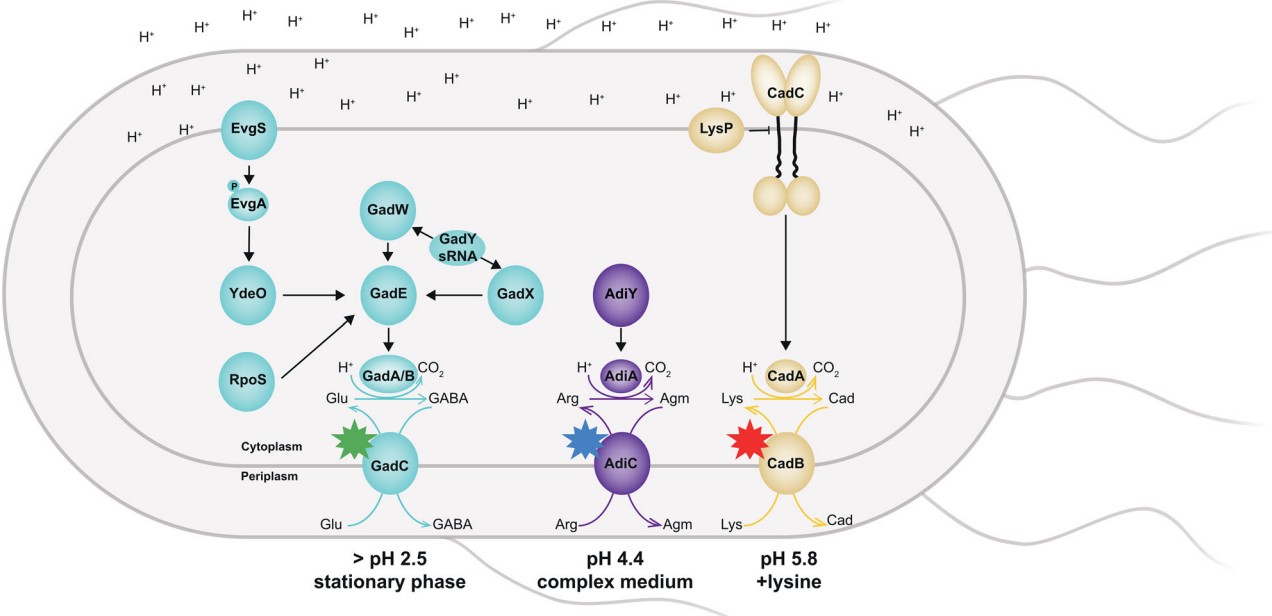

**Fig. 1 The regulatory network of inducible amino acid decarboxylase-antiporter systems in _E. coli_ three-color strain.** Regulated induction of the Gad (blue symbols), Adi (purple symbols), and Cad (yellow symbols) systems requires a network of membrane-integrated pH-sensors, namely EvgS and CadC; interconnected transcription factors; and the sRNA GadY. The glutamate decarboxylases GadA and GadB convert glutamate (Glu) into γ-aminobutyrate (GABA), which is excreted by GadC. The arginine decarboxylase AdiA converts arginine (Arg) into agmatine (Agm), which is excreted by AdiC. The lysine decarboxylase CadA converts lysine (Lys) into cadaverine (Cad), which is excreted by CadB. Adapted from[3, 15, 24, 25, 47, 74]. The stimuli leading to an induction of each of the three AR systems is indicated below each system. Each fluorescent hybrid of the three antiporter is indicated with a star symbol: GadC:eGFP (green star), AdiC:mCerulean (blue star), and CadB:mCherry (red star) according to the hybrid with the fluorophore eGFP, mCerulean, and mCherry, respectively.

points, cells were imaged, and the fluorescence intensities were quantified (Fig. 2b).

GadC:eGFP showed the highest production during the stationary phase and under the acidic conditions of pH 5.8 and pH 4.4 (Fig. 2b–d, $t_{300}$). In the stationary phase at physiological pH, GadC:eGFP was produced two times less than under acidic conditions; however, its heterogenous distribution increased, as indicated by noise values of 0.11 (defined as the standard deviation divided by the mean of log-transformed intensity values[42]) ($t_{300}$) (Fig. 2c; left panel). When cells were exposed to mild acid stress (pH 5.8), GadC:eGFP was homogenously distributed among individual cells in the _E. coli_ population and exhibited low noise values of 0.04–0.08 (Fig. 2b, c; left panel).

The Cad system is activated by mild acidic stress in the presence of lysine; therefore, CadB:mCherry became visible after 2.5 h of growth in a minimal medium supplemented with lysine at pH 5.8. At this time point, CadB:mCherry was heterogeneously distributed, as reflected by a high noise value of 0.37 (Fig. 2b–d), which is in perfect agreement with previous experiments of CadB tagged with eGFP[41]. The distribution of CadB:mCherry remained heterogeneous at pH 4.4 in complex medium, but the mean value increased 1.5-fold (Fig. 2c; right panel, $t_{300}$).

The Adi system is only activated under stronger acidic conditions, such as pH 4.4 in a complex medium with a noise value of 0.12 (Fig. 2b, c; middle panel).

Overall, the distribution of CadB:mCherry among single _E. coli_ cells under acid stress was not a symmetric Gaussian-like distribution but rather an asymmetric right-skewed distribution (Fig. 2d), as up to 83% of the population produced CadB:mCherry at varying high levels, and the remaining cells were in the OFF state, which was in agreement with previous findings using CadB:eGFP[41]. The distribution of GadC:eGFP and AdiC:mCerulean follows a symmetric Gaussian-like distribution (Fig. 2d),

which indicated that individual cells produce different amounts of GadC:eGFP and AdiC:mCerulean. However, almost all cells produced varying amounts of GadC:eGFP (99%), whereas only half of the population (47.7%) produced AdiC:mCerulean at pH 4.4 (Fig. 2d; right panel). As a control, non-tagged _E. coli_ MG1655 cells were examined under the same conditions; these cells exhibited low background fluorescence and extremely low noise values (Supplementary Table 1).

To exclude pH-effects on the fluorophores, each of the three fluorophores (eGFP, mCherry, and mCerulean) was fused to the antiporter gene _gadC_, because GadC is produced under all tested external pH values. Neither the different fluorophores nor the external pH affected the output or noise of the fluorescent hybrids (Supplementary Fig. 2; Supplementary Table 1).

Overall, the three inducible AR systems were heterogeneously activated with a different degree of noise. The Cad system showed the highest heterogeneous distribution (pH 5.8 $t_{150}$) of all three systems, whereas the Gad and Adi systems showed comparable noise values (0.11 and 0.12, respectively), but under different conditions. The Gad system was the most heterogenous during the stationary phase, whereas the Adi system was activated heterogeneously by stronger acid stress (pH 4.4).

**Simultaneous activation of the AR systems in single _E. coli_ cells.** Since the three AR systems showed a heterogenous output, although each had a different strength of heterogeneity, we examined their simultaneous activation within individual _E. coli_ cells under exposure to consecutively increasing acid stress (Fig. 2a).

The correlation between GadC:eGFP and CadB:mCherry was analyzed at the time point of the highest degree of heterogeneity of the Cad system (pH 5.8 $t_{150}$; Fig. 3a); however, no correlation occurred, thus indicating that there was a lack of dependency

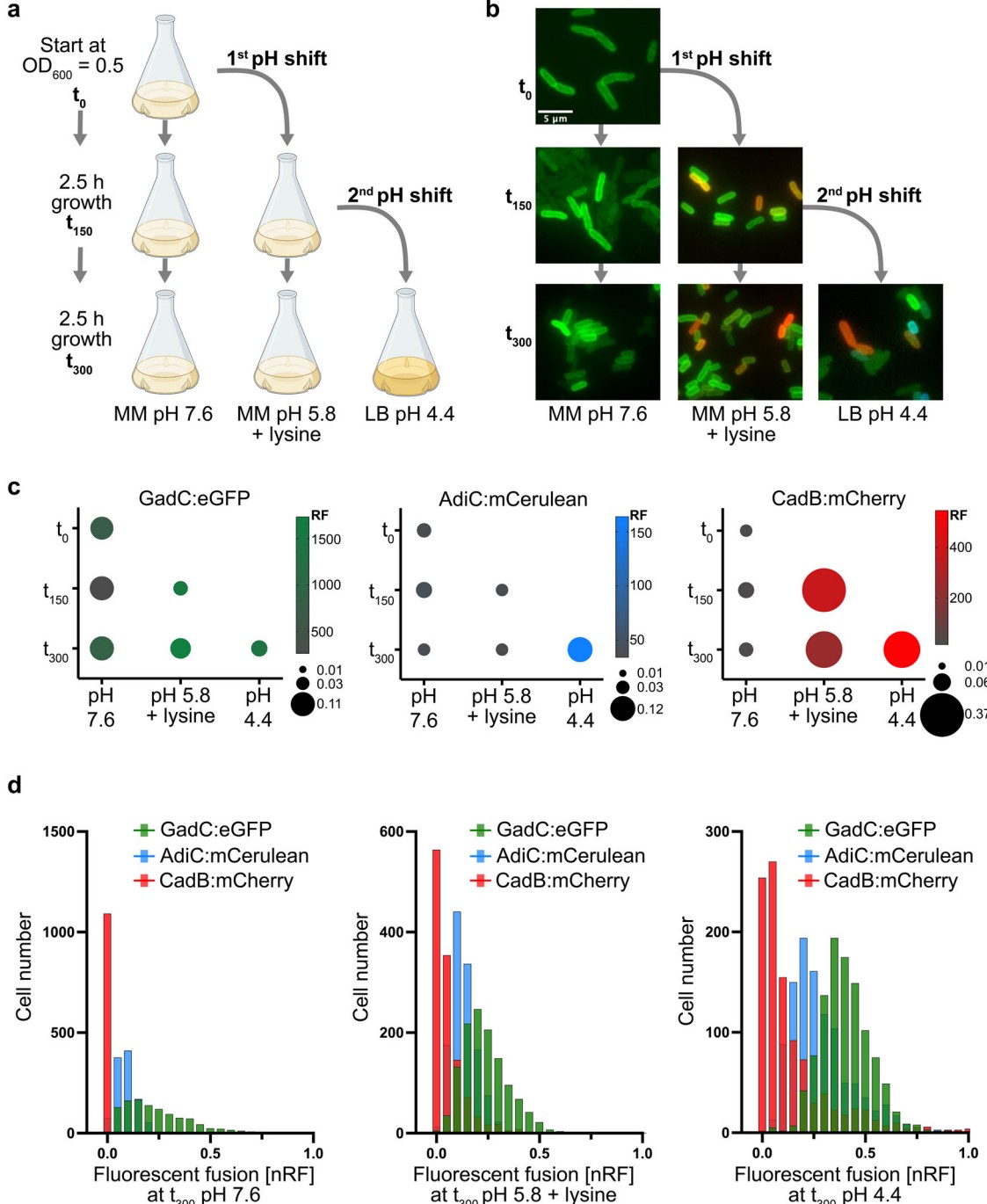

**Fig. 2 Heterogenous activation of the three inducible AR systems in *E. coli* in response to acid stress. a** Schematic representation of the experimental setup: at the exponential phase ($t_0$), the three-color reporter strain *E. coli gadC:eGFP-adiC:mCerulean-cadB:mCherry* was shifted from the MM at pH 7.6 to MM at pH 5.8 supplemented with 10 mM lysine, followed by a second shift to the LB medium at pH 4.4. In parallel, half of the culture was incubated under the original pH condition. **b** Fluorescent microscopic images of the three-color reporter strain *E. coli gadC:eGFP-adiC:mCerulean-cadB:mCherry*, cultivated as presented in (**a**), were taken at the indicated time points. Representative fluorescence overlay images are shown. Scale bar, 5 μm. **c** Quantified noise and mean RF were calculated for 1000 cells per condition and time point of the cultivated three-color reporter strain *E. coli gadC:eGFP-adiC:mCerulean-cadB:mCherry*, as presented in (**a**). Noise (standard deviation/mean of log-transformed values) is presented by the size of the dots (the higher the noise, the larger the size of the dot), and the average of the RF is presented with a color code for each fluorescent hybrid, GadC:eGFP (left panel), AdiC:mCerulean (middle panel), and CadB:mCherry (right panel). RF was quantified by using the MicrobeJ plugin of the ImageJ software of the fluorescent microscopic images. **d** Histogram presentation of the nRF quantified for 1000 cells per fluorescent hybrid, GadC:eGFP, AdiC:mCerulean, and CadB:mCherry, grown at $t_{300}$ in MM pH 7.6 (left panel), MM at pH 5.8 supplemented with 10 mM lysine (middle panel), and LB medium at pH 4.4 (right panel), respectively. A comparison of the frequencies of CadB:mCherry with AdiC:mCerulean and with GadC:eGFP using the Chi-square test showed a *p*-value < 0.0001 for each time point. LB lysogeny broth, MM minimal medium, nRF normalized relative fluorescence intensity, RF relative fluorescence intensity.

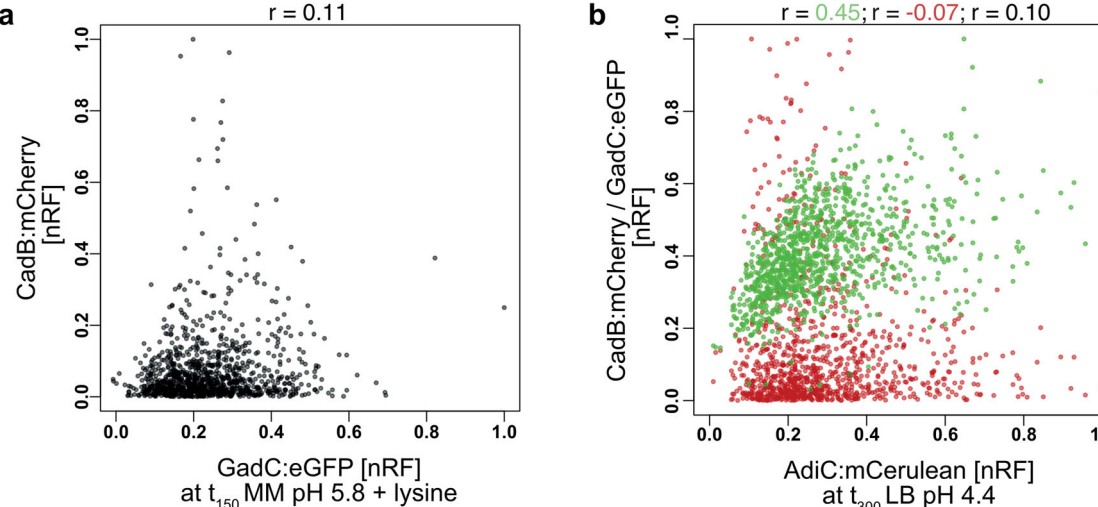

**Fig. 3 Simultaneous activation of the AR systems in single *E. coli*. a** Normalized fluorescence intensity (nRF) presented in a correlation plot of CadC:eGFP versus CadB:mCherry of the strain *E. coli gadC:eGFP-adiC.mCerulean-cadB:mCherry* in MM at pH 5.8 supplemented with lysine at $t_{150}$. Pearson's correlation coefficient ($r$) is shown on top of the graph: $r = 0.11$ for GadC:eGFP and CadB:mCherry with a *p*-value of $3.28e^{-4}$. **b** nRF presented in a correlation plot of AdiC:mCerulean versus GadC:eGFP (green dots) and versus CadB:mCherry (red dots) of *E. coli gadC:eGFP-adiC.mCerulean-cadB:mCherry* in LB medium at pH 4.4 at $t_{300}$. Pearson's correlation coefficient is shown on top of the graph: $r = 0.45$ for GadC:eGFP and AdiC:mCerulean (green) with a p-value of $4.4^{-56}$; $r = -0.07$ for AdiC:mCerulean and CadB:mCherry (red) with a *p*-value of $<0.05$; $r = 0.10$ for GadC:eGFP and CadB:mCherry (black) with a *p*-value of $<0.05$. Pearson's correlation coefficient was calculated using GraphPad Prism 9.1.0. Correlation plots were created using R 4.0.3. LB lysogeny broth, MM minimal medium, RF relative fluorescence intensity, nRF normalized RF values, noise standard deviation/mean of log-transformed values.

between these two systems (Pearson's correlation coefficient, $r = 0.11$). During the stationary phase under strong acid stress (pH 4.4 $t_{300}$), when all three systems were activated, GadC:eGFP and AdiC:mCerulean showed a positive correlation ($r = 0.45$; green dots); however, no correlation occurred between CadB:mCherry and GadC:eGFP or AdiC:mCerulean ($r = 0.10$ and $r = -0.07$, respectively) (Fig. 3b). In summary, in stationary phase under strong acid stress, almost all cells activated the Gad system; however, simultaneous activation of the Cad and Adi systems did not occur.

**Phylogenetic distribution of the components of the three AR systems within the bacterial kingdom.** To identify connecting regulators between the three AR systems and to understand the segregation of the Adi and the Cad systems in *E. coli*, we used a bioinformatic approach to investigate the presence and distribution of regulatory components of the three AR systems within the bacterial kingdom. We used the antiporters GadC, AdiC, and CadB as the basis for the construction of phylogenetic trees and focused on discovering a potential co-occurrence of their specific regulators (Fig. 1; Supplementary Table 2).

According to our study, there are 1112 homologs of *E. coli* GadC that mainly occur in the phylum Proteobacteria (42.4%), such as *Enterobacteriaceae* (31.1%, with 27.8% belonging to *Escherichia*); *Morganellaceae* (4.5%); and *Yersiniaceae* (3.0%). Moreover, 51.3% of the GadC homologs belong to the bacteria of Firmicutes (56.0%), such as *Listeriaceae* (14.8%), *Enterococcaceae* (14.1%), *Clostridaceae* (9.8%), *Lactobacillaceae* (7.3%) and *Streptococcaceae* (5.2%) (Fig. 4a). Although GadC and GadB homologs are widely occurring, the regulators responsible for the sophisticated regulation of the Gad system of *E. coli* are mainly present in *Enterobacteriaceae* (Fig. 4a). GadW, GadX, GadY, GadE, and YdeO are predominantly conserved in *Escherichia* (89.3%) and, to some extent, in *Shigella* (0.04%) species. GadE is found in *Escherichia* and *Shigella* but also in other *Enterobacteriaceae* species, such as *Citrobacter freundii* and *Kluyvera ascorbate*. (Fig. 4a; Supplementary Table 2). The carbon induced

starvation transcriptional regulator CsiR, a member of the GntR-family, mainly occurs within γ-proteobacteria in 30.3% of species containing a GadC homolog, especially in *Enterobacteriaceae* (94.9%), such as *E. coli* (Fig. 4a). In *E. coli*, CsiR presumably affects all three AR systems: overexpression of CsiR stimulates binding to promoters of the Gad system, *gadX* and *gadW*; of the Cad system, *cadC*; and leads to lower expression of the *adiY* and *adiA* genes[16]. The sensor kinase EvgS is mainly found in *Enterobacteriaceae* (85.6%) but also in members of *Hafniaceae*, like *Hafnia alvei*, and *Yersiniaceae*, such as *Yersinia fredericksenii* and *Serratia fonticola*, all of which belong to γ-proteobacteria (Fig. 4a).

The components of the Adi and Cad systems are mainly conserved within γ-proteobacteria. In total, 756 AdiC homologs were identified within *Enterobacteriaceae* (70.8%), *Lysobacteraceae* (11.6%), *Yersiniaceae* (7.5%), and *Morganellaceae* (4.4%) (Fig. 4b). Of species containing an AdiC homolog, the Adi system-specific transcriptional activator AdiY is present in 69.0%, all of which, like *Escherichia* (68.9%), *Salmonella* (20.1%), and *Shigella* (4.9%), belong to the *Enterobacteriaceae* family. CsiR is distributed in a similar manner and is present in 70.4% of species containing an AdiC homolog. The 533 CsiR homologs that were identified mainly belonged to *Enterobacteriaceae* and *Hafniaceae* species, such as *Escherichia* (67.7%), *Salmonella* (19.7%), *Shigella* (4.9%), and *Hafnia* (2.8%) species. Moreover, CsiR homologs co-occur to a high degree with AdiY homologs, as 96.4% of species containing AdiY also have a CsiR homolog, except for *Hafina* species, which lack AdiY homologs (Fig. 4b; Supplementary Table 2).

The antiporter CadB of the Cad system was found mainly in γ-proteobacteria. 857 homologs were identified belonging to *Enterobacteriaceae* (55.8%), *Vibrionaceae* (20.0%), *Aeromonadaceae* (7.0%), *Hafniaceae* (5.6%), and *Yersiniaceae* (4.7%). CadB mainly co-occurs with CadA and CadC homologs. Of the species that have a CadB homolog, 94.3% also possess a CadA homolog of the cognate lysine decarboxylase of the Cad system, and 95.5% also have a homolog of the cognate pH-sensor CadC (Fig. 4c). In total, 573 homologs of the lysine-permease LysP were identified,

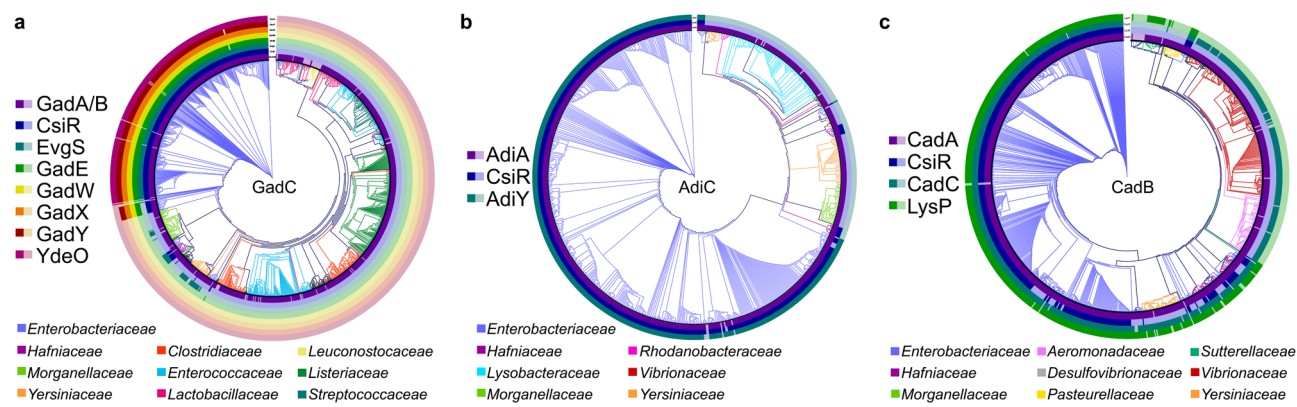

**Fig. 4 Phylogenetic trees of the *E. coli* antiporters GadC, AdiC, and CadB and co-occurring regulators of the three AR systems within the bacterial kingdom. a** The protein sequences of 1112 of *E. coli* GadC homologs were aligned, and a phylogenetic tree was generated, which is shown as a circular cladogram. The branches of the tree were colored according to the family of the organisms containing a GadC homolog. The presence of different regulatory components CsiR, EvgS, GadE, GadW, GadX, GadY, and YdeO and the decarboxylases GadA and GadB (GadA/B) is indicated by solid colors in the respective ring layer. Translucent colors represent components that were not detected. **b** The protein sequences of 756 *E. coli* AdiC homologs were aligned, and a phylogenetic tree was generated, which is shown as a circular cladogram. The branches of the tree are colored according to the family of organisms containing an AdiC homolog. The presence of the different regulatory components CsiR and AdiY as well as the decarboxylase AdiA is indicated by solid colors in the respective ring layer. If these components were not found, the colors are translucent. **c** The protein sequences of 857 *E. coli* CadB homologs were aligned, and a phylogenetic tree was generated, which is shown as a circular cladogram. The branches of the tree are colored according to the family of organisms containing a CadB homolog. The presence of the different regulatory components CsiR, CadC, and LysP as well as the decarboxylase CadA are indicated by solid colors in the respective ring layer. Translucent colors represent components that were not detected. Identifiers of all GadC, AdiC, and CadB homologs with an indication of the presence or absence of their main regulators are summarized in Supplementary Data 1.

66.9% of which were found in species containing a CadB homolog. As previously reported, LysP predominantly co-occurs with the Cad system within the *Enterobacteriaceae* family (82.3%) and is mostly absent in *Vibrionaceae* (Fig. 4c; Supplementary Table 2)[41]. CsiR homologs are less coupled with CadB compared to other components of the Cad system. However, similar to LysP, CsiR is mainly found in *Enterobacteriaceae* and *Hafniaceae*, such as *Escherichia* (49.1%), *Klebsiella* (23.4%), *Salmonella* (14.6%), and *Hafnia* (4.1%) species. In summary, the regulatory components specific to *E. coli* of the three AR systems that were studied were predominantly conserved in *Enterobacteriaceae*. The Gad system was the most widespread AR system, whereas the Adi and Cad systems were restricted to γ-proteobacteria. Interestingly, the decarboxylases, together with their cognate antiporters, were more widely distributed than other regulatory components. The central regulators of the Cad system and the Adi system were more specific to the *Enterobacteriaceae* family.

**Mechanisms of heterogenous activation of the three AR systems.** To gain insight into the molecular mechanisms of heterogeneous activation of the AR systems in *E. coli*, we analyzed the importance of different transcriptional regulators. We focused mainly on the Adi and Cad systems, as these systems exhibited a higher degree of heterogeneity. First, we assessed the role of the regulator CsiR, which co-occurs with the components of the Gad, Adi, and Cad systems, mainly in *Enterobacteriaceae* (Fig. 4), and is supposed to modulate AR in *E. coli*[16]. However, artificially increased CsiR levels affected only the Adi system and not the Gad or Cad systems (Supplementary Table 1). Higher intracellular CsiR levels caused a 2.4-fold decrease in the amount of AdiC:mCerulean but did not affect the heterogenous distribution. The measured noise value was 0.10 under these conditions and is consistent with the wild type (Supplementary Table 1). These results are in good agreement with the suggested repressing impact of CsiR on the Adi system according to ChiP-Seq analysis[16].

Second, we already know that the native low CadC copy number generates heterogenous activation of the Cad system, as

an increase in the CadC copy number correlates with a decrease in heterogeneous distribution of the Cad system[41]. In addition, CadC itself is distributed heterogeneously[43]. Because of the mutually exclusive activation of the Cad and Adi systems (Fig. 3b), we tested whether CadC also has an influence on the Adi system. When we artificially elevated the copy number of CadC, we found four-times lower mean fluorescence of AdiC:mCherry (Fig. 5a). Moreover, the heterogenous distribution of AdiC:mCherry was much lower under this condition (Fig. 5a). It should be noted that the higher level of CadC decreased the amount of GadC:mCerulean by only two-fold in cells exposed to pH 5.8, and the already less heterogenous distribution of the Gad system was not affected (noise value of 0.05, which is comparable to that of the wild type) (Supplementary Table 1). In agreement with previous observations, the higher level of CadC increased the production of CadB:eGFP while reducing its heterogeneous distribution (Supplementary Table 1).

Third, AdiY also belongs to low copy number regulators with approximately 11–37 copies per cell[44]. Based on our previous experience with CadC[41], we analyzed the influence of the copy number of AdiY on heterogenous activation of the Adi system in pH 4.4-stressed cells. We increased the copy number by placing *adiY* under the control of the P$_{BAD}$ promoter on a plasmid. The artificially increased copy number of AdiY resulted in a two-fold increase in the mean fluorescence and thus a higher amount of AdiC:mCerulean, which is not surprising, since AdiY is the main transcriptional activator of *adiC*. Furthermore, the heterogeneous distribution of AdiC:mCerulean was strongly reduced (Fig. 5b). However, *E. coli* cells producing a higher copy number of AdiY had a growth disadvantage under acid stress compared with wild-type cells (Fig. 5c). A similar effect on growth was previously demonstrated for *E. coli* cells producing a higher CadC copy number[41] (Supplementary Fig. 3).

The effect of CadC on the Adi system could be direct or indirect. A direct effect, i.e., that CadC binds upstream of *adiC* as a transcriptional regulator, can be excluded, as we did not find a second binding site for CadC in *E. coli*[43,45,46]. Therefore, we hypothesized that an indirect effect was caused by a CadC-

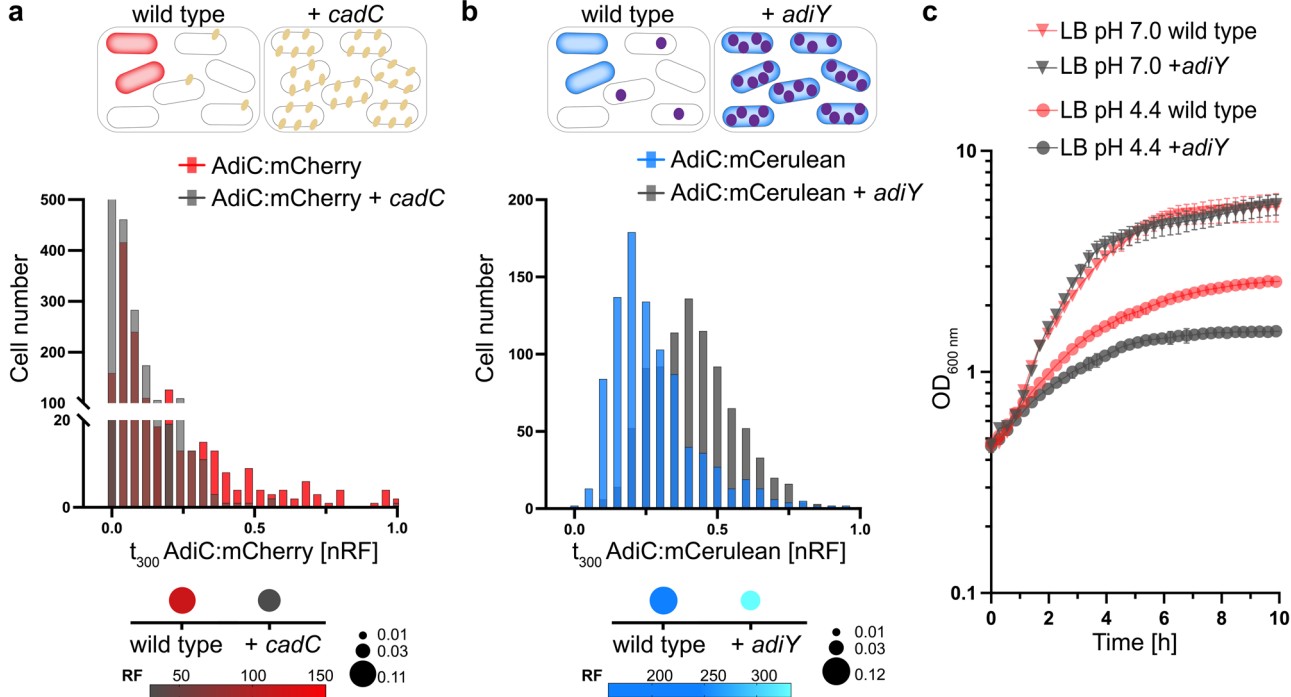

**Fig. 5 An increase of the copy number of AdiY or CadC affects the degree of heterogeneity of the Adi system. a** nRF values of cells expressing AdiC:mCherry in the wild type background (red) or with an elevated CadC copy number (gray) at pH 4.4 in LB medium at $t_{300}$. +cadC, the expression of cadC under the control of the arabinose (0.1%)-inducible promoter in plasmid pBAD24. The comparison of frequencies of AdiC:mCherry in wild type or +cadC cells using the Chi-square test showed a p-value < 0.001. On top of the histogram: red cells represent the AdiC:mCherry ON state; +cadC is represented by yellow dots. Below the histogram: noise is presented by the size of the dots; average RF is represented by color intensity (Supplementary Table 1). RF was quantified using the MicrobeJ plugin of the ImageJ software of the fluorescent microscopic images for 1000 cells per condition. nRF normalized RF values. **b** nRF values of cells expressing AdiC:mCerulean in the wild type background (blue) or with an elevated AdiY copy number (gray) and cultivated as in (a). + adiY, expression of adiY under the control of the arabinose (0.1%)-inducible promoter in plasmid pBAD24. Comparison of the frequencies of AdiC:mCerulean in wild type or +adiY cells using the Chi-square test showed a p-value < 0.0001. On top of the histogram: blue cells represent the AdiC:mCerulean ON state; +adiY is represented by purple dots. Below the histogram: noise level is represented by the size of the dots; average RF is represented by color intensity (Supplementary Table 1). RF was quantified using the MicrobeJ plugin of the ImageJ software of the fluorescent microscopic images for 1000 cells per condition. nRF normalized RF values. **c** E. coli MG1655 wild type transformed with the plasmid pBAD24-adiY or with empty pBAD24 were grown in minimal medium at pH 5.8 supplemented with lysine and then shifted to LB medium of the indicated pH values. Growth ($OD_{600}$) was determined every 10 min in microtiter plates with continuous shaking. All experiments were performed three times (n = 3), and error bars represent standard deviation of the means. LB lysogeny broth.

induced increase in $H^+$-consuming lysine decarboxylase CadA, thus conferring moderate AR to all cells (homogeneous response; Fig. 6a). To test this hypothesis, we deleted cadA from our reporter strain (ΔcadA gadC:eGFP-adiC:mCerulean-cadB:mCherry). Deletion of cadA reduced the heterogenous distribution of AdiC:mCerulean and strongly increased the mean fluorescence of AdiC:mCerulean (by 11-fold), thus suggesting that many more cells produced AdiC:mCerulean compared with the wild type (Fig. 6a). Deletion of cadA did not affect the heterogenous distribution of either GadC:eGFP or CadB:mCherry in pH-4.4-stressed cells. Only the overall mean level of CadB:mCherry was increased under this condition, which was due to the lack of negative feedback inhibition of cadaverine on CadC (Supplementary Table 1)[34,36].

To further support these results, we analyzed AdiY-driven pH-dependent activation of the adiA promoter at the population level. In wild-type cells, adiA expression began when cells were exposed to an acidic environment of pH < 4.8 (Fig. 6b). The increased number of CadC molecules did not change the pH-dependent induction profile but caused downregulation of the adiA promoter activity. In contrast, in the cadA mutant, the promoter activity of adiA not only increased, but the onset of induction shifted to a higher external pH such as 5.8 (Fig. 6b). These data suggest that activation of the Adi system is influenced

by the intracellular pH, as a manipulation of the cytosolic proton concentration by either deletion of cadA or overproduction of cadA (triggered by increased CadC copy numbers) leads to increased activation or downregulation, respectively, of the Adi system in E. coli.

## Discussion

To counteract acidic environments, many bacteria possess inducible AR systems that rely on $H^+$-consuming amino acid decarboxylases and their corresponding antiporters. The latter function as importers for corresponding amino acids and exporters for decarboxylated products. The number and complexity of the three inducible Gad, Adi, and Cad AR systems vary among bacteria and reflect an adaptation to the needs of their individual natural habitat[47]. Our phylogenetic analysis of the distribution of antiporters GadC, AdiC, and CadB revealed that the Gad system is the most widely distributed AR system, whereas the Adi system and Cad system are restricted to γ-proteobacteria (Fig. 4). Consistent with previous reports, Salmonella has Adi and Cad systems, whereas Shigella has the Gad system[47]. In Shigella, we also identified the Adi system, and in some species, the Cad system, i.e., in S. boydii and S. flexneri (Supplementary Table 2, Supplementary Data 1). However, not only does the number of

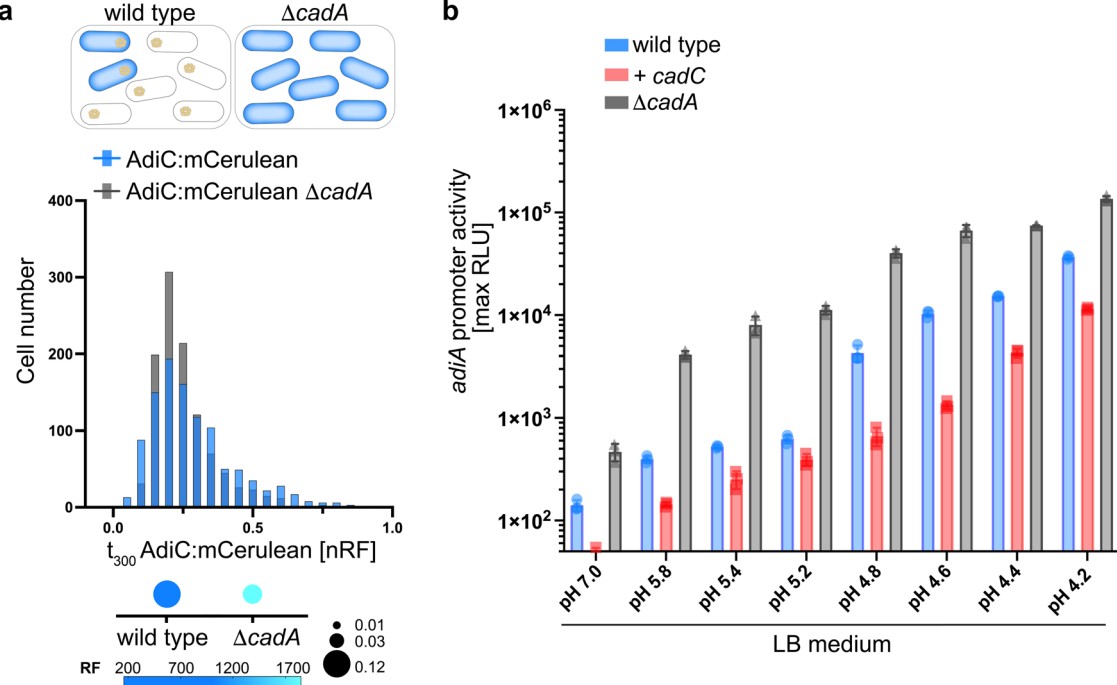

**Fig. 6 Manipulation of intracellular stress resistance by either eliminating (ΔcadA) or increasing (+cadC) the activity of the Cad system affects the degree of heterogeneity and pH-dependent induction of the Adi system. a** Normalized RF values of cells expressing AdiC:mCerulean in the *E. coli* wild type background (blue) or the *cadA* mutant (gray). Cells were monitored after growth at pH 4.4 in LB medium at $t_{300}$. On top of the histogram, the cells are shown in a schematic overview, with blue cells representing the AdiC:mCerulean ON state and yellow dots representing CadA. Below the histogram, the noise (standard deviation/mean of log-transformed values) is presented by the size of the dots (the higher the noise, the larger the size of the dot), and the average RF is represented by a blue color (the more intense the blue, the higher the average relative fluorescence intensity). RF was quantified using the MicrobeJ plugin of the ImageJ software of the fluorescent microscopic images for 1000 cells per condition. The calculated mean RF and noise values are summarized in Supplementary Table 1. nRF normalized RF values. **b** *E. coli* MG1655 wild type and the *cadA* mutant (each transformed with the plasmid pBBR1-MCS5-P*adiA*-lux) were grown in MM at pH 5.8 supplemented with 10 mM lysine and then shifted to LB medium of the indicated pH values. Luminescence and growth were determined every 10 min in microtiter plates with a Tecan Infinite F500 system (Tecan, Crailsheim, Germany). Data are reported as relative light units (RLUs) in counts per second per milliliter per $OD_{600}$, and maximal RLU at 1.9 h is shown. All experiments were performed three times ($n = 3$), and error bars represent standard deviation of the means. Growth over 10 h is presented in Supplementary Fig. 3 at selected pH values. LB lysogeny broth, MM minimal medium, RF relative fluorescence intensity.

AR systems possessed by bacteria differ, but the complexity of their regulation varies (Fig. 4). The most sophisticated regulatory network of AR systems is found in *E. coli*. Its close relatives *E. albertii* have almost all components as well but lack the EvgSA two-component system and YdeO.

For the first time, we investigated the activation of the three major AR systems in *E. coli* simultaneously at the single-cell level under consecutive increasing acid stress. By using the three-color reporter strain *E. coli* gadC:eGFP-adiC:mCerulean-cadB:mCherry, we confirmed that each of the three AR systems is specialized for a certain strength of acid stress under certain environmental conditions. The Gad system was activated under mild acid strength and during the stationary phase. The Cad system was induced at pH 5.8 and required the presence of lysine. The Adi system was induced at pH 4.4 and required the lysogeny broth medium, which contains tryptone and yeast extract. However, activation of the three systems differs greatly from cell to cell (Figs. 2 and 7). The Cad system showed the highest heterogeneous distribution of all three systems, when activated at pH 5.8 in the presence of lysine. Under almost all conditions, all cells of the population activate the Gad system but to varying degrees. The Adi system is activated heterogeneously under strong acid stress (pH 4.4) (Fig. 2).

Phenotypic variations can be beneficial for the *E. coli* population under acid stress. Our results provide a model of how the three inducible AR systems overlap and generate a division of labor and functional cooperation in the population (Fig. 7). All cells of the *E. coli* population individually adapt to mild acid stress by activating the Gad system to varying degrees, and the glutamate decarboxylases GadA and GadB might initially utilize intracellularly available glutamate under the consumption of protons. As described below, the antiporter GadC becomes more important under extreme acid stress. It should also be noted here that the decarboxylase GadB also undergoes a pH-dependent conformational change and exhibits an activity optimum at low pH[48]. Furthermore, glutamate is the most abundant intracellular metabolite, with a concentration of 100 mM, and its concentration is much higher compared to that of lysine (0.41 mM) and arginine (0.57 mM)[49]. Under stronger acid stress (pH 5.8 to pH 4.4), some cells in the population activate the Cad system, whereas others activate the Adi system. These two subpopulations each contribute to acid stress relief by secreting the more alkaline cadaverine and agmatine, respectively (Fig. 7). These two polyamines are considered common goods and contribute to an increase of the extracellular pH, which benefits the whole population. This behavior can be considered as an example of true division of labor in bacteria, as the metabolic burden of the individual cells to produce both the Adi and Cad systems is prevented, but nevertheless, the whole population benefits from the elevation of external pH by secretion of cadaverine and agmatine (Fig. 7). The fact that homogenous production of the components of the Adi system in all cells would be a burden for

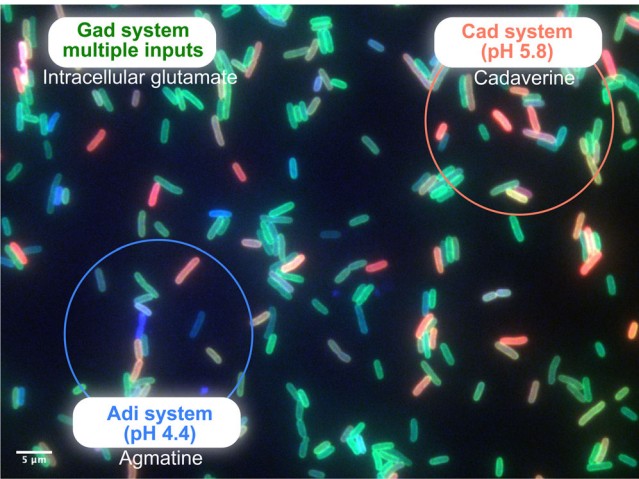

**Fig. 7 Model of the heterogenous activation of three inducible acid resistance systems and division of labor in the *E. coli* population.** Fluorescent microscopic images of the three-color reporter strain *E. coli gadC:eGFP-adiC:mCerulean-cadB:mCherry* at pH 4.4 in a complex medium at t$_{300}$. All cells of the *E. coli* population individually adapt to acid stress by activating the Gad system to varying degrees (green fluorescent cells) due to multiple extracellular and intracellular inputs. These cells utilize primarily intracellularly available glutamate under consumption of protons to increase their internal pH. Under stronger acid stress (pH 5.8 to pH 4.4), some cells in the population activate the Cad system (red fluorescent cells), while others activate the Adi system (blue fluorescent cells). These cells secrete (as indicated by the rings) the more alkaline products cadaverine and agmatine, respectively, thereby contributing to acid-stress relief with an increase of the extracellular pH, which benefits the whole population. In addition, their internal pH is elevated by consuming protons due to the conversion of lysine to cadaverine and arginine to agmatine, respectively.

the population is clearly shown by the reduced growth rate under this condition (Fig. 6c), which was similar for a homogenous production of the Cad system[41] (Supplementary Fig. 3).

We were then interested in the molecular mechanisms that not only lead to this phenotypic heterogeneity but that are also heritable. CadC and AdiY are low copy number proteins; therefore, they are stochastically distributed among the cells of the population[44]. Here and in our previous studies[41], we showed that an increase of their copy number was sufficient to shift the population into homogeneous behavior. Cells that activated the Cad system are protected against acid stress due to the H$^+$-consuming activity of the CadA decarboxylase. It should be noted that CadA is produced on average with 60,000 copies per cell, which accounts for at least 2% of all cytoplasmic proteins in CadA-producing cells[50]. In cells lacking the Cad system, the intracellular pH decreases under strong external acid stress, such as pH 4.4. Our results have shown that any manipulation of the activity of the Cad system, such as its elimination (ΔcadA) or increase (+cadC), affected the degree of heterogeneity and pH-dependent induction of the Adi system, which suggests that the regulator of the Adi system is an intracellular pH sensor that senses a decrease of cytoplasmic pH.

Induction of the Gad system of *E. coli* is controlled by an unusually high number of transcription factors, the stationary sigma factor, and a small RNA GadY (Figs. 1 and 4a). Moreover, its activation is strongly influenced by the medium composition and growth phase[10,17]. The transcription factors GadX and GadW were already associated with higher noise levels of their target genes of unstressed cells[42] and potentially contribute to the heterogenous distribution of GadC:eGFP (Fig. 2b). In addition, the stationary sigma factor RpoS is known generate extensive

transcriptional heterogeneity[51–53]. Thus, the Gad system integrates multiple extracellular and intracellular inputs, which results in quantitative differences and continuous variation of phenotypic traits in the individual cells of the population. The noisy activation of the *gadBC* promoter enables prediction of the survival of single cells[54]. Usually, high levels of noise are found for genes that are involved in either the energy metabolism of carbon sources or adaptation to stress, like osmotic pressure, temperature extremes, starvation response, pH response, and mechanical, nutritional, or oxidative stress. Indeed, the Gad system is a major system under extreme acid stress (pH ≤ 2.5)[10]. The antiporter GadC preferentially transports protonated glutamate (Glu$^+$) in exchange for protonated GABA (GABA$^+$) at an external pH of 3.0 and lower[21–23]. This transport activity is crucial for protecting cells against incoming protons. It should be noted that at extremely low pH values, such as pH 2.5, only <0.01% of a non-preadapted population survives[27], and therefore, for maintenance of the population, it is extremely important that all *E. coli* cells produce the Gad system, albeit with different levels.

We can only speculate to what extent this model might hold up in a natural epitope of *E. coli*, such as digestive system of humans or animals. Before *E. coli* arrives in the intestine, it is exposed to an extreme acid stress with HCl in the stomach that this bacterium can only survive with the help of the Gad system. The mean pH in the proximal small intestine is 6.6 and increases in the terminal ileum to 7.5. Then, there is a sharp fall in pH to 6.4 in the caecum, and an increase in colon with a final value of 7.0[55]. Depending on the availability of the amino acids lysine and arginine, the Cad and Adi systems might be switched on under these conditions. The latter system is increasingly activated due to the anaerobic conditions[31]. Both systems will still be heterogeneously distributed due to the low copy number of the main regulators CadC and AdiY. Fermentation products, such as acetate, also activate the Gad system[56]. The extent to which microcolonies of *E. coli* exist in the colon, in which common goods such as cadaverine and agmatine can change the micromilieu, is also unclear, as is the extent to which the host plays a role in acid stress. In fact, the levels of putrescine, a conversion product of agmatine, and cadaverine were found to be 15- and 3-fold, increased, respectively, in the inflamed mouse intestine mono-colonized with *E. coli* compared to an equally colonized healthy mouse cohort[57].

Considered together, our results provide a model for the advantage of *E. coli* to have three AR systems that allow for division of labor in the population and ensure its survival over a wide range of low pH values, thus making this bacterium highly acid-stress resistant.

## Methods

**Bacteria and growth conditions.** Bacterial strains and plasmids used in this study are listed in Table 1. *E. coli* strains were cultivated in LB medium (10 g/liter NaCl, 10 g/l tryptone, 5 g/l yeast extract) or in Kim-Epstein (KE) minimal medium[58] adjusted to pH 5.8 (MM pH 5.8) or pH 7.6 (MM pH 7.6), using the corresponding phosphate buffer. *E. coli* strains were incubated aerobically in a rotary shaker at 37 °C. KE medium was always supplemented with 0.2% (w/v) glucose. When indicated lysine was added to a final concentration of 10 mM.

If necessary, media were supplemented with 100 µg/ml ampicillin, or 50 µg/ml kanamycin sulfate. To allow the growth of the conjugation strain *E. coli* WM3064, *meso*-diamino-pimelic acid (DAP) was added to a final concentration of 300 µM.

**Construction of plasmids.** Molecular methods were carried out according to standard protocols or according to the manufacturer's instructions. Kits for the isolation of plasmids and the purification of PCR products were purchased from Süd-Laborbedarf (SLG; Gauting, Germany). Enzymes and HiFi DNA Assembly Master Mix were purchased from New England BioLabs (Frankfurt, Germany).

To construct different fluorescent fusions with *gadC*, *adiC* and *cadB*, to be inserted in-frame chromosomally, the corresponding flanking regions (600 bp upstream and downstream) of *gadC*, *adiC* or *cadB* were amplified by PCR using MG1655 genomic DNA as template. Plasmid pET-*mCherry-cadC*[43] was used as

**Table 1 Strains and plasmids used in this study.**

| Strains | Relevant genotype or description | Reference |
|---|---|---|
| *E. coli* MG1655 | K-12 F⁻ λ⁻ *ilvG⁻ rfb-50 rph-1* | [68] |
| *E. coli* DH5αλpir | *endA1 hsdR17 glnV44* (= *supE44*) *thi-1 recA1 gyrA96 relA1* φ80′*lac*Δ(*lacZ*)M15 Δ(*lacZYA-argF*)U169 *zdg*−232::Tn10 *uidA*::*pir* + | [69] |
| *E. coli* WM3064 | *thrB1004 pro thi rpsL hsdS lacZ*ΔM15 RP4-1360 Δ(*araBAD*)567 Δ*dapA1341*::[*erm pir*] | W. Metcalf, Univ. of Illinois, Urbana |
| *E. coli* MG1655 *cadB:egfp* | Chromosomally integrated C-terminal *cadB:egfp* fusion in *E. coli* MG1655 | [41] |
| *E. coli* MG1655 *adiC:mCherry-cadB:egfp* | Chromosomally integrated C-terminal *adiC:mCherry* and *cadB:egfp* fusion in *E. coli* MG1655 | This work |
| *E. coli* MG1655 *gadC:mCerulean-adiC:mCherry-cadB:egfp* | Chromosomally integrated C-terminal *gadC:mCerulean*ᵃ, *adiC:mCherry* and *cadB:egfp* fusion in *E. coli* MG1655 | This work |
| *E. coli* MG1655 *adiC:mCerulean-cadB:egfp* | Chromosomally integrated C-terminal *adiC:mCerulean* and *cadB:egfp* fusion in *E. coli* MG1655 | This work |
| *E. coli* MG1655 *gadC:mCherry-adiC:mCerulean-cadB:egfp* | Chromosomally integrated C-terminal *gadC:mCherry*ᵃ, *adiC:mCerulean* and *cadB:egfp* fusion in *E. coli* MG1655 | This work |
| *E. coli* MG1655 *cadB:mCherry* | Chromosomally integrated C-terminal *cadB:mCherry* fusion in *E. coli* MG1655 | This work |
| *E. coli* MG1655 *adiC:mCerulean-cadB:mCherry* | Chromosomally integrated C-terminal *adiC:mCerulean* and *cadB:mCherry* fusion in *E. coli* MG1655 | This work |
| *E. coli* MG1655 *gadC:egfp-adiC:mCerulean-cadB:mCherry* | Chromosomally integrated C-terminal *gadC:egfp*ᵃ, *adiC:mCerulean* and *cadB:mCherry* fusion in *E. coli* MG1655 | This work |
| *E. coli* MG1655 Δ*cadA gadC:eGFP-adiC:mCerulean-cadB:mCherry* | In-frame deletion of *cadA* in *E. coli* MG1655 *gadC:egfp*ᵃ-*adiC:mCerulean-cadB:mCherry* | This work |
| *E. coli* Δ*gadC* | Deletion of *gadC* (JW1487), *gadC*::Km | [70] |
| *E. coli* Δ*adiC* | Deletion of *adiC* (JW4076), *adiC*::Km | [70] |
| *E. coli* Δ*cadA* | Deletion of *cadA* (JW4092), *cadA*::Km | [70] |
| Plasmids | | |
| pBAD24 | Arabinose-inducible P$_{BAD}$ promoter, pBR322 ori, Amp$^R$ | [71] |
| pBAD-*cadC* | *cadC* under control of arabinose inducible promoter in pBAD24, Amp$^R$ | [40] |
| pNTPS138-R6KT | *mobRP4* + *ori*-R6K *sacB*; suicide plasmid for in-frame deletions, Km$^R$ | [72] |
| pNPTS138-R6KT-*cadB:egfp*-EC | pNPTS-138-R6KT-derived suicide plasmid for in-frame insertion of *cadB:egfp* in *E. coli* MG1655, Km$^R$ | [41] |
| pNPTS138-R6KT-*gadC:egfp* | pNPTS-138-R6KT-derived suicide plasmid for in-frame insertion of *gadC:egfp*ᵃ in *E. coli* MG1655, Km$^R$ | This work |
| pNPTS138-R6KT-*cadB:mCherry* | pNPTS-138-R6KT-derived suicide plasmid for in-frame insertion of *cadB:mCherry* in *E. coli* MG1655, Km$^R$ | This work |
| pNPTS138-R6KT-*adiC:mCerulean* | pNPTS-138-R6KT-derived suicide plasmid for in-frame insertion of *adiC:mCerulean* in *E. coli* MG1655 strains, Km$^R$ | This work |
| pNPTS138-R6KT-*gadC:mCherry* | pNPTS-138-R6KT-derived suicide plasmid for in-frame insertion of *gadC:mCherry*ᵃ in *E. coli* MG1655, Km$^R$ | This work |
| pNPTS138-R6KT-*gadC:mCerulean* | pNPTS-138-R6KT-derived suicide plasmid for in-frame insertion of *gadC:mCerulean*ᵃ in *E. coli* MG1655, Km$^R$ | This work |
| pNPTS138-R6KT-*adiC:mCherry* | pNPTS-138-R6KT-derived suicide plasmid for in-frame insertion of *adiC:mCherry* in *E. coli* MG1655, Km$^R$ | This work |
| pBAD-His$_6$-*csiR* | N-terminal His$_6$-tagged *csiR* in pBAD24, Amp$^R$ | This work |
| pBAD-His$_6$-*adiY* | N-terminal His$_6$-tagged *adiY* in pBAD24, Amp$^R$ | This work |
| pBBR1-MCS5-TT-RBS-*lux* | *luxCDABE* and terminators lambda T0 *rrnB1* T1 cloned into pBBR1-MCS5 for plasmid-based transcriptional fusions, Gm$^R$ | [73] |
| pBBR1-MCS5-P$_{adiA}$-*lux* | *adiA* promoter controlling expression of *luxCDABE*, in pBBR1-MCS5-TT-RBS-*lux*, Gm$^R$ | This work |
| pNPTS138-R6KT-Δ*cadA* | pNPTS-138-R6KT-derived suicide plasmid for in-frame deletion of *cadA* in MG1655 *gadC:eGFP*ᵃ-*adiC:mCerulean-cadB:mCherry*, Km$^R$ | This work |
| pET-*mCherry-cadC* | N-terminal fusion of CadC with mCherry in pET16b, Amp$^R$ | [43] |
| pK18mob2-TriFluoR | pK18mob2 Km carrying triple reporter construct with MCS I-*cerulean*, PT5-*mCherry*, MCS II-*mVenus* | [59] |

*EC E. coli* MG1655.
ᵃAll *gadC* genes encode a truncated protein (amino acids 1–470) that lacks the C-plug[21].

template (720 bp) to amplify *mCherry*. Plasmid pK18mob2-TriFluoR[59] was used as template (720 bp) to amplify *mCherry*. Plasmid pNPTS138-R6KT-*cadB:egfp*-EC was used as template (720 bp) to amplify *egfp*. After purification of the different fragments, the respective combinations were assembled via Gibson assembly[60] into EcoRV-digested pNPTS138-R6KT plasmid, resulting in the plasmids pNPTS138-R6KT-*gadC:egfp*, pNPTS138-R6KT-*gadC:mCherry*, pNPTS138-R6KT-*gadC:mCerulean*, pNPTS138-R6KT-*cadB:mCherry*, pNPTS138-R6KT-*adiC:mCerulean* and pNPTS138-R6KT-*adiC:mCherry*. Each plasmid was verified by colony PCR and sequencing.

Functional hybrid proteins of GadC with each of the three different fluorophores was only achieved by using a shorter version of GadC (amino acids 1–470), in which the C-terminal C-plug is removed. Truncation of this C-plug does not affect the transport activity, but shifts the pH-optimum to a higher pH[21].

To construct a marker-less in-frame deletion of *cadA* in *E. coli* MG1655 *gadC:eGFP-adiC:mCerulean-cadB:mCherry*, the suicide plasmid pNPTS138-R6KT-Δ*cadA* was generated. Briefly, flanking regions (1000 bp upstream and downstream) of *cadA* were amplified by PCR using MG1655 *gadC:eGFP-adiC:mCerulean-cadB:mCherry* genomic DNA as template. After purification of the different fragments, the DNA fragments were assembled via Gibson assembly[60] into by PCR

linearized pNPTS138-R6KT plasmid, resulting in the plasmid pNPTS138-R6KT-ΔcadA. Correct insertion was verified by sequence analysis using primer pNTPS_Seq_fwd.

For construction of the reporter plasmid pBBR1-MCS5-P_{adiY}-lux, 200 bp of the region upstream of adiA was amplified by PCR using primers (PadiA_XbaI_fwd and PadiA_XmaI_rev) and MG1655 genomic DNA as template, and cloned into plasmid pBBR1-MCS5-TT-RBS-lux using restriction sites XbaI and XmaI. Correct insertion was verified by colony PCR and sequencing.

For construction of N-terminal His_6-tagged AdiY and His_6-tagged CsiR, adiY and csiR were amplified by PCR using MG1655 genomic DNA as template. The codons for the His_6-tag were introduced by using a respective forward primer. After purification of the different fragments, the DNA fragments were assembled via Gibson assembly[60] into SmaI-digested pBAD24 plasmid, resulting in the plasmids pBAD24-His_6-adiY and pBAD24-His_6-csiR. Correct insertion was verified by colony PCR and sequencing.

**Construction of chromosomally integrated fluorescent fusions and deletion strains.** The genes encoding the three different fluorophores, egfp, mCherry and mCerulean, are separately C-terminally fused to the genes gadC, adiC and cadB encoding the antiporter of the three AR systems. Expression of each fusion is under the control of the respective native promoter.

To generate the three-color E. coli MG1655 gadC:mCerulean-adiC:mCherry-cadB:egfp strain, at first the suicide plasmid pNPTS138-R6KT-adiC:mCherry was introduced into E. coli MG1655 cadB:egfp by conjugative mating using E. coli WM3064 as a donor in LB medium containing DAP. Single-crossover integration mutants were selected on LB plates containing kanamycin but lacking DAP. Single colonies were grown over a day without antibiotics and plated on LB plates containing 10% (w/v) sucrose but lacking NaCl to select for plasmid excision. Kanamycin-sensitive colonies were checked for targeted deletion by colony PCR using primers bracketing the site of the insertion. Insertion of mCherry was verified by colony PCR and sequencing resulting in the strain E. coli MG1655 adiC:mCherry-cadB:egfp. In the next step tagging of gadC with mCerulean was achieved using the suicide plasmid pNPTS138-R6KT-gadC:mCerulean. The plasmid pNPTS138-R6KT-gadC:mCerulean was introduced into E. coli MG1655 adiC:mCherry-cadB:egfp by conjugative mating using E. coli WM3064 as a donor in LB medium containing DAP as described above. Insertion of mCerulean was verified by colony PCR and sequencing, resulting in the strain E. coli MG1655 gadC:mCerulean-adiC:mCherry-cadB:egfp.

To generate the three-color E. coli MG1655 gadC:mCherry-adiC:mCerulean-cadB:egfp strain, at first the suicide plasmid pNPTS138-R6KT-adiC:mCerulean was introduced into E. coli MG1655 cadB:egfp by conjugative mating using E. coli WM3064 as a donor in LB medium containing DAP as described above. Insertion of mCerulean was verified by colony PCR and sequencing, resulting in the strain E. coli MG1655 adiC:mCerulean-cadB:egfp. In the next step tagging of gadC with mCherry was achieved using the suicide plasmid pNPTS138-R6KT-gadC:mCherry. The plasmid pNPTS138-R6KT-gadC:mCherry was introduced into E. coli MG1655 adiC:mCerulean-cadB:egfp by conjugative mating using E. coli WM3064 as a donor in LB medium containing DAP as described above. Insertion of mCherry was verified by colony PCR and sequencing resulting in the strain E. coli MG1655 gadC:mCherry-adiC:mCerulean-cadB:egfp.

To generate the two-color E. coli MG1655 gadC:mCerulean-cadB:mCherry strain, at first the suicide plasmid pNPTS138-R6KT-cadB:mCherry was introduced into E. coli MG1655 by conjugative mating using E. coli WM3064 as a donor in LB medium containing DAP as described above. Insertion of mCherry was verified by colony PCR and sequencing, resulting in the strain E. coli MG1655 cadB:mCherry. In the next step tagging of gadC with mCerulean in this strain was achieved using the suicide plasmid pNPTS138-R6KT-gadC:mCerulean. The plasmid pNPTS138-R6KT-gadC:mCerulean was introduced into E. coli MG1655 cadB:mCherry by conjugative mating using E. coli WM3064 as a donor in LB medium containing DAP as described above. Insertion of mCerulean was verified by colony PCR and sequencing resulting in the strain E. coli MG1655 gadC:mCerulean-cadB:mCherry.

To generate the three-color E. coli MG1655 gadC:egfp-adiC:mCerulean-cadB:mCherry strain, at first the suicide plasmid pNPTS138-R6KT-adiC:mCerulean was introduced into E. coli MG1655 cadB:mCherry by conjugative mating using E. coli WM3064 as a donor in LB medium containing DAP as described above. Insertion of mCerulean was verified by colony PCR and sequencing, resulting in the strain E. coli MG1655 adiC:mCerulean-cadB:mCherry. In the next step tagging of gadC with egfp was achieved using the suicide plasmid pNPTS138-R6KT-gadC:egfp. The plasmid pNPTS138-R6KT-gadC:egfp was introduced into E. coli MG1655 adiC:mCerulean-cadB:mCherry by conjugative mating using E. coli WM3064 as a donor in LB medium containing DAP as described above. Insertion of egfp was verified by colony PCR and sequencing, resulting in the strain E. coli MG1655 gadC:egfp-adiC:mCerulean-cadB:mCherry.

Construction of the marker-less in-frame deletion strain of cadA in E. coli MG1655 gadC:eGFP-adiC:mCerulean-cadB:mCherry was achieved using the suicide plasmid pNPTS138-R6KT-ΔcadA. The plasmid pNPTS138-R6KT-ΔcadA was introduced into E. coli MG1655 gadC:eGFP-adiC:mCerulean-cadB:mCherry by conjugative mating using E. coli WM3064 as a donor in LB medium containing DAP as described above. Deletion of cadA was verified by colony PCR and sequencing, resulting in the strain E. coli MG1655 ΔcadA gadC:eGFP-adiC:mCerulean-cadB:mCherry.

**In vivo fluorescence microscopy and data analysis.** To analyze the spatial temporal localization of the different fluorescent hybrids in E. coli, overnight cultures were prepared in KE medium pH 7.6 and aerobically cultivated at 37 °C. The overnight cultures were used to inoculate day cultures (OD_{600} of 0.1) in fresh medium at pH 7.6. At an OD_{600} of 0.5 (t_0), half of the culture was gently centrifuged and resuspended in KE medium pH 5.8 as the first pH shift. The rest of the culture continued growing in KE medium pH 7.6 at 37 °C. Then the cultures were cultivated aerobically at 37 °C for 2.5 h (t_{150}), and half of the culture was gently centrifuged and resuspended in LB medium pH 4.4 as the second pH shift. The rest of the culture continued growing in KE medium pH 5.8 + lysine at 37 °C (t_{300}). At the beginning of the experiment (t_0) and at every pH shift (t_{150} and t_{300}), 2 µl of the culture was spotted on 1% (w/v) agarose pads (prepared with the respective media), placed onto microscope slides and covered with a coverslip. Subsequently, images were taken on a Leica DMi8 inverted microscope equipped with a Leica DFC365 FX camera (Wetzlar, Germany). An excitation wavelength of 546 nm and a 605-nm emission filter with a 75-nm bandwidth was used for mCherry fluorescence with an exposure of 500 ms, gain 5, and 100% intensity for mCherry-tagged strains. An excitation wavelength of 485 nm and a 510-nm emission filter with a 75-nm bandwidth was used for eGFP fluorescence with an exposure of 500 ms, gain 5, and 100% intensity for eGFP-tagged strains. An excitation wavelength of 436 nm and 480-nm emission filter with a 40-nm bandwidth was used for mCerulean fluorescence with an exposure of 500 ms, gain 5, and 100% intensity for mCerulean-tagged strains.

To analyze the influence of an increased copy number of CsiR, the E. coli strains gadC:mCerulean-cadB:mCherry and adiC:mCerulean-cadB:mCherry were transformed with plasmid pBAD-His_6-csiR by electroporation. 0.1% (w/v) L-arabinose was added throughout the experiment and cells were cultivated as described above.

To analyze the influence of an increased copy number of AdiY, the E. coli strain adiC:mCerulean-cadB:mCherry was transformed with plasmid pBAD-His_6-adiY by electroporation. 0.1% (w/v) L-arabinose was added throughout the experiment and cells were cultivated as described above.

To analyze the influence of an increased copy number of CadC, the E. coli strains gadC:mCerulean-cadB:mCherry and adiC:mCherry-cadB:eGFP were transformed with plasmid pBAD-His_6-adiY by electroporation. 0.1% (w/v) L-arabinose was added throughout the experiment and cells were cultivated as described above.

To quantify relative fluorescent intensities (RF) representing fluorescent hybrids of single cells, phase contrast and fluorescent images were analyzed using the ImageJ[61] plugin MicrobeJ[62]. Default settings of MicrobeJ was used for cell segmentation (Fit shape, rod-shaped bacteria) apart from the following settings: area: 0.1-max µm²; length: 1.2–5 µm; width: 0.1–1 µm; curvature 0.−0.15 and angularity 0.−0.25 for E. coli cells. In total >1000 cells were quantified per strain and condition and time point. The background of the agarose pad was subtracted from each cell per field of view.

Mean and standard deviation (std) of the relative fluorescence (RF) were quantified using MicrobeJ and background was subtracted. Moreover, the RF of the GFP channel was corrected for the overlapping signal of the CFP channel via linear regression with the experimentally determined proportionality factor of 0.456. The noise was defined as std/mean of log-transformed values per sample. Log-transformed values were used because the coefficient of variation, as a measure of noise, is based on the assumption that the underlying data are normally distributed. However, not all data were normally distributed under all conditions, therefore the log-transformed values were used (log(x + 1)) as previously published for CadB:eGFP[41]. In addition, this helped to overcome the problem that the noise becomes higher with lower gene expression, which is often observed[42].

In order to compare fluorescence intensity values of the different fluorophores, the RF values were normalized (nRF) according to the highest values, as each of the three fluorophores has a different brightness (eGFP: 33; mCerulean: 16; mCherry: 15)[63].

Statistical analysis and presentation were performed using GraphPad Prism 9.1.0 and R 4.0.3[64].

**Localization of fluorescently tagged GadC, AdiC, and CadB hybrids.** To verify the localization of fluorescently tagged GadC, AdiC and CadB in the membrane, the three-color E. coli strains gadC:egfp-adiC:mCerulean-cadB:mCherry, gadC:mCherry-adiC:mCerulean-cadB:eGFP and gadC:mCerulean-adiC:mCherry-cadB:eGFP were cultivated as described above for fluorescence microscopy. After the second pH shift to pH 4.4 (t_{300}) cells were harvested and then adjusted to OD_{600} = 30. Cells were disrupted by passage through a high-pressure cell disrupter (Constant Systems, Northants, United Kingdom) in ice-cold disruption buffer (50 mM Tris-HCl pH 7.5, 10% (v/v) glycerol, 10 mM MgCl_2, 100 mM NaCl, 1 mM dithiothreitol, 0.5 mM PMSF and 0.03 mg ml⁻¹ DNase). After removal of intact cells and cell debris via centrifugation (5000 × g, 30 min, 4 °C), membrane vesicles were collected by ultracentrifugation (45,000 × g, 60 min, 4 °C), whereas the pellet contained the membrane fraction and the supernatant the cytoplasm. The membrane fractions were separated by SDS-PAGE[65] on 12.5% acrylamide gels and transferred to a nitrocellulose membrane. Fluorophore-tagged proteins were labeled either with the primary polyclonal α-mCherry antibody (Invitrogen), α-GFP antibody (Roche) or the α-mCerulean antibody PABG1 (Chromotek). The α-

rabbit or α-mouse alkaline phosphatase-conjugated antibody (Rockland Immunochemicals) was used as secondary antibody according to the manufacturer's recommendations. Localization of the secondary antibody was visualized using colorimetric detection of alkaline phosphatase activity with 5-bromo-4-chloro-3-indolyl phosphate and nitro blue tetrazolium chloride. As ladder the PageRuler Prestained Protein Ladder (10–180 kDa, Thermo Fisher) was used.

**Functionality of fluorescent antiporter hybrids.** The functionality of the Cad-B:eGFP hybrid was previously confirmed using a liquid-based colorimetric assay with a pH indicator[41]. To test the functionality of the fluorescent hybrid proteins GadC:eGFP and AdiC:mCerulean, an acid survival assay was performed as described previously[10,41] with the following modifications. E. coli strains were cultivated in LB medium (pH 7.6) to an $OD_{600}$ of 0.6–0.8, then the cultures were adjusted to an $OD_{600}$ of 0.5 and resuspended in LB medium with a pH of 3.0 or 4.4. The low pH challenge was conducted at 37 °C and samples were collected immediately after resuspension ($t = 0$) and then hourly for 3 h. Samples were serially diluted and plated onto LB agar plates to assess the number of colonies surviving the acid challenge. As controls, the parental E. coli MG1655 and the gadC and adiC deletion mutants were used.

**Measurement of adiA promoter activity in vivo.** In vivo promoter activity of adiA was probed with a luminescence-based reporter ($P_{adiA}$-luxCDABE). The influence of the Cad system and the AdiY copy number was tested in cells exposed to different external pH values in LB medium using luminescence as readout. The strains E. coli MG1655 carrying pBBR1-MCS5-$P_{adiA}$-lux, E. coli MG1655 ΔcadA carrying pBBR1-MCS5-$P_{adiA}$-lux, E. coli MG1655 carrying pBBR1-MCS5-$P_{adiA}$-lux co-transformed with pBAD-cadC were incubated in KE medium pH 7.6 supplemented with the respective antibiotics overnight. The overnight cultures were inoculated to an $OD_{600}$ of 0.1 in fresh KE medium pH 7.6, aerobically cultivated until exponential phase and then shifted to KE medium pH 5.8 + 10 mM lysine and cultivated for another 2.5 h at 37 °C (comparable to the experimental setup described in Fig. 2a). In the next step, the cultures were shifted into a 96-well plate and aerobically cultivated at 37 °C in LB medium at different pH values supplemented with the respective antibiotics. Bioluminescence and growth were determined every 10 min in the microtiter plates with a Tecan Infinite F500 system (Tecan, Crailsheim, Germany). Data are reported as relative light units (RLU) in counts per second per milliliter per $OD_{600}$.

To analyze the effect of an increased copy number of CadC or AdiY, 0.1% (w/v) L-arabinose was added during the growth at pH 5.8 + 10 mM lysine and at pH 4.4. Thereby the CadC copy number is elevated to about 100 CadC molecules per cell, whereas the E. coli MG1655 wild type strain contains only ≤4 CadC molecules per cell[40].

**Alignment and construction of phylogenetic trees.** To identify non-redundant orthologues of the components of the three AR systems in relation to each antiporter GadC, AdiC and CadB, a Protein BLAST search of the NCBI RefSeq protein database[66] using the full-length sequence of either GadC (XasA), AdiC and CadB from E. coli MG1655 as the query sequence was carried out. Different expect thresholds were used for the Protein BLAST search (e < $10^{-100}$ for GadC, e < $10^{-120}$ for AdiC and CadB), in order to include all homologous yet functional similar proteins (June 2020). A tolerance of 10% of the amino acid length was set as default parameters.

Three alignments and phylogenetic trees were constructed, each based on the antiporter GadC, AdiC or CadB with the different components, and visualized as metadata. Therefore, a pairwise alignment of 1112 sequences for GadC, of 755 sequences for AdiC and of 857 sequences for CadB was done with a progressive algorithm from the software CLC Main Workbench 20.0.3 (CLC Bio Qiagen, Hilden, Germany) using the following parameters: gap open cost 10, gap extension cost 1, high accuracy[67]. The results served as the basis for the construction of three phylogenetic trees by the software's high-accuracy, distance-based neighbor-joining algorithm (100 bootstrap replicates and the Jukes-Cantor distance correction as default parameters). The branch lengths, therefore, represent the degree of evolutionary divergence between any two nodes in the tree.

We screened the organisms containing a GadC orthologue for the presence of the GadB, GadE, GadX, GadW, GadY, CsiR, EvgS and YdeO by searching for orthologues of E. coli MG1655 with NCBI Protein BLAST (e < $10^{-20}$ for GadE, e < $10^{-50}$ for CsiR, e < $10^{-80}$ for YdeO, e < $10^{-100}$ for GadB, e < $10^{-100}$ for GadW and GadX, e < $10^{-170}$ for EvgS). Homologs of the sRNA GadY were determined by a NCBI Nucleotide BLAST[66] search using gadY from E. coli MG1655 as the query sequence with an expect threshold of e < $10^{-20}$.

We screened the organisms containing a AdiC orthologue for the presence of the AdiA, AdiY and CsiR by searching for orthologues of E. coli MG1655 with NCBI Protein BLAST (e < $10^{-170}$ for AdiA, e < $10^{-100}$ for AdiY, e < $10^{-50}$ for CsiR).

We screened the organisms containing an CadB orthologue for the presence of the CadA, CadC, LysP and CsiR by searching for orthologues of E. coli MG1655 with NCBI Protein BLAST (e < $10^{-170}$ for CadA, e < $10^{-120}$ for LysP, e < $10^{-50}$ for CsiR). For the phylogenetic distribution of CadC, the data from our previous publication[41] were used.

All data for the construction of these phylogenetic trees are available in Supplementary Data 1.

**Statistics and reproducibility.** All experiments were repeated multiple (three replicates unless stated) times to ensure reproducibility. Statistics were performed using either a two-tailed t-test, one-way ANOVA, or a two-tailed Pearson correlation coefficients comparison using GraphPad Prism 9.1.0. All graphs were plotted using the GraphPad Prism 9.1.0 and R 4.0.3[64]. For multi-well plate assays, replicates were also contained within the plate to determine well to well variability.

**Reporting summary.** Further information on research design is available in the Nature Research Reporting Summary linked to this article.

## Data availability

Supplementary Data 1 contains a list of all GadC, AdiC, and CadB homologs identified by a local alignment search based on the full-length sequence of E. coli GadC, AdiC, and CadB underlying Fig. 4. All source data underlying the graphs and charts presented in Figs. 2, 3, 5 and 6 are presented in Supplementary Data 2. Plasmids and primer sequences are available on request.

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

## Acknowledgements

We thank Jürgen Lassak for stimulating scientific discussions. We thank Sabine Peschek for excellent technical assistance and Anke Becker for providing the plasmid pK18mob2-TriFluoR. This work was financially supported by the Deutsche Forschungsgemeinschaft [Projects 26942323 (TRR174), 395357507 (SFB 1371) and 471254198 (JU270/21-1) to K.J.].

## Author contributions

K.J. and S.B. designed the study. S.B. and K.S. performed the experiments and analyzed the data. S.K. performed the phylogenetic analysis. S.B. and K.J. wrote the manuscript with input from all authors.

## Funding

## Competing interests

The authors declare no competing interests.
