## [Peer Review File · Communications Biology]

Reviewers' comments:

Reviewer #1 (Remarks to the Author):

This is a well-written paper which examines an interesting question about induction of AR systems in E coli at the single cell level. Previous work done in the senior author's lab on the Cad system behaviour at single cell level places them very well to do this sort of work. It represents a considerable amount of work both in terms of strain construction and data gathering and analysis. For the most part I am happy with the methods used, the quality of the data gathered, and the interpretation thereof, with just a few points that the authors could comment on (listed below).

1. RcsB is also an important additional component in the regulation of the GAD system, having effects both on GadE and EvgA, and should be included in the list of known regulators.
2. I like the way of presenting the data in the histograms as it shows the overlapping distributions; however I'd suggest adding borders to the bars, as sometimes it is a little tricky to see the different values when different colours are superimposed.
3. They hypothesise an indirect effect causes the correlation between the Cad and Adi systems, and the data with the cadA deletion seems to support this. But would a similar effect not be expected due to the effect of expression of GadA and GadB, which also depleted intracellular protons?
4. Related to this (and possibly explaining it) I am not sure of the statement (line 448/449) about the Gad decarboxylases using intracellular glutamate. The activity of the Gad system at low pH (2.5) requires external glutamate and an active GadC, despite the high internal glutamate levels. The authors may be referring to activity at an external pH of 5.8; however the pH optimum of GadA/B is quite a bit lower than this (in fact this might be the answer to the point (3) above: there may be very limited GadAB activity in the experimental conditions used here). Note that EvgS will be inactive after the second pH shift (as 4.4 is below its activity range, and EvgS is in any case not activated even at pH 5.8 in LB).
5. For clarity, I'd suggest rephrasing lines 454-455 to "...cadaverine or agmatine respectively". Similarly, I'd rephrase lines 494-495 to read "...only <0.01% of the population survives if the Gad system has not been pre-induced".
6. The "division of labour" model is interesting and fits the data well. The authors should however clarify how relevant this is in vivo where it is most likely to have evolved. In the gut, conditions will be very different to those described here - it will micro-aerobic or anaerobic, pH shock will be provided mostly by organic acids which have very different properties to inorganic ones (at least once cells have passed the stomach, which is far more acidic than the conditions described here), and E coli cells will be a small minority of a complex microbial population, so the effects of secreted molecules may be minor. This is not to invalidate their model, but its limitations in understanding the behaviour of E coli in its true habitat should be made clearer.
7. A minor point, but I don't find the data shown in Figure S1 panel B for adiC to be very convincing. There's no difference in cell counts between the 10⁻⁶ and 10⁻⁷ dilutions for adiC::mCerulean, and for a system where the effect is relatively subtle on pH sensitivity, I think plates with a larger number of colonies on should be used (ie a whole plate for each dilution, rather than a small spot). However, I don't think this alters the validity of the rest of the paper.

Reviewer #2 (Remarks to the Author):

The authors describe how E. coli's three main acid-dependent amino-acid decarboxylation systems are induced under successive ranges of acidic pH. The triple fluorescent strain generates elegant data, although the essential phenomenon of these acid responses has been known for a while.

The authors do an excellent job of recounting known information in the Introduction and crediting

appropriate sources. They are to be commended for not trying to claim they are the first to observed events reported back in the 90s.

What they do appear to show is the real-time kinetics of differential expression of the three systems. The heterogeneity of cell response is also shown. The authors might be interested to check Martinez et al 2012, which reports heterogeneity in cytoplasmic pH response to external acid shift: doi.org/10.1128/AEM.00354-12

One concern of this approach is that a cell with three fluorescent reporters may behave somewhat differently from the wild-type. However, the authors have some controls for these possibilities.

Since the Gad system appears the most heterogeneous, the authors should comment on RpoS regulation which known to generate extensive genetic and transcriptional heterogeneity.

For the figure legends: All the Y axes should be log scale; this would avoid the awkward breaks, and would be more appropriate for regulatory data, in which fold-change is generally the measure of on-off circuits.

Reviewer #3 (Remarks to the Author):

Brameyer et al used a triple reporter strain to characterise cell individuality in the activation of three distinct responses to acid treatment in *E. coli*, namely glutamate decarboxylase, arginine decarboxylase, and lysine decarboxylase. They convincingly show that possessing all three systems confer an advantage to *E. coli* (and possibly several other species) that is therefore able to survive over a wide pH range. The manuscript is very well written and nicely presented with some beautiful fluorescence images (Figure 7 is superb!), the data are very robust.

Therefore, I believe that this manuscript should be published in *Communications Biology* after some minor adjustments as detailed below.

Line 35 please define AR

Lines 43-44: in the case of *E. coli* the intracellular pH does indeed change during standard growth in LB, please see <https://www.frontiersin.org/articles/10.3389/fmicb.2018.01739/full> (and references therein) and distinct *E. coli* subpopulations display different pH ranges possibly due to differential expression of the three AR systems reported in this manuscript, please see <https://journals.asm.org/doi/full/10.1128/mBio.00909-21> (and references therein)

Lines 63-64: I found this sentence unclear because of the contrast between "mild acidic" and "extremely low", perhaps rephrase?

Lines 126: could the authors please show data about the (lack of) spectra overlap between the three selected fluorophores

Line 135: this very nicely agrees with previous transcriptomic data <https://www.frontiersin.org/articles/10.3389/fmicb.2018.01739/full> (and references therein)

Lines 201-210 and Fig. 3: could the authors please add statistical significance to corroborate their Pearson's correlation coefficients estimates? This is, for example, a built-in function in software such as prism graphpad

Point-to-point answers to the Reviewer comments (in blue):

Re: COMMSBIO-21-2988-T (Division of labor and collective functionality in *Escherichia coli* under acid stress)

Reviewer #1 (Remarks to the Author):

This is a well-written paper which examines an interesting question about induction of AR systems in *E. coli* at the single cell level. Previous work done in the senior author's lab on the Cad system behaviour at single cell level places them very well to do this sort of work. It represents a considerable amount of work both in terms of strain construction and data gathering and analysis. For the most part I am happy with the methods used, the quality of the data gathered, and the interpretation thereof, with just a few points that the authors could comment on (listed below).

1. RcsB is also an important additional component in the regulation of the GAD system, having effects both on GadE and EvgA, and should be included in the list of known regulators.

We completely agree and inserted the following sentence including three references in the revised manuscript:

“In addition, RcsB is a critical partner of GadE and the binding of both regulators as a heterodimer to the GAD box activates *gadA* transcription.” (Castanié-Cornet et al., 2010; Johnson et al., 2011; Krin et al., 2010). (Lines 81-83).

2. I like the way of presenting the data in the histograms as it shows the overlapping distributions; however I'd suggest adding borders to the bars, as sometimes it is a little tricky to see the different values when different colours are superimposed.

Thank you for this suggestion, we revised the figures and inserted grey borders to the bars of the histograms in figures 2d, 5a, 5b and 6a. See below Figure 5 as an example (all figures are integrated in the revised manuscript).

3. They hypothesise an indirect effect causes the correlation between the Cad and Adi systems, and the data with the *cadA* deletion seems to support this. But would a similar

effect not be expected due to the effect of expression of GadA and GadB, which also depleted intracellular protons?

We completely agree and stated this already in the manuscript that “All cells of the *E. coli* population individually adapt to mild acid stress by activating the Gad system to varying degrees, and the glutamate decarboxylases GadA and GadB ... decarboxylate glutamate under the consumption of protons.” (Lines 454 ff.). Since we did not find a correlation between the Gad and Cad systems, but a mutually exclusive induction of the Cad and Adi systems in cells exposed to a pH value of 4.4, we were interested in the molecular mechanisms of this phenomenon. Therefore, we tested, among other things, the effect of CadA overproduction. The overproduction of GadA and GadB could also downregulate the activation of the Adi system. However, due to the different pH values required for the onset of activation (Gad system pH 7.6 + stationary phase versus Adi system pH 4.4), we believe this would be a rather unphysiological setup.

4. Related to this (and possibly explaining it) I am not sure of the statement (line 448/449) about the Gad decarboxylases using intracellular glutamate. The activity of the Gad system at low pH (2.5) requires external glutamate and an active GadC, despite the high internal glutamate levels. The authors may be referring to activity at an external pH of 5.8; however the pH optimum of GadA/B is quite a bit lower than this (in fact this might be the answer to the point (3) above: there may be very limited GadAB activity in the experimental conditions used here). Note that EvgS will be inactive after the second pH shift (as 4.4 is below its activity range, and EvgS is in any case not activated even at pH 5.8 in LB).

It was not our intention to say that external glutamate does not play a role. We completely agree that the activity of Gad system requires external glutamate to enable cells to survive at extreme acid stress. Our argument only concerned the functionality of the Gad system of *E. coli* cells exposed to mild acid stress or in stationary phase.

According to the thoughtful suggestions of the reviewer we revised our statements as follows (additions underlined, ~~deletions crossed out~~):

1. Lines 72-74: “It is important to note that the activity of GadC is pH-dependent, and the antiporter preferentially exchanges protonated glutamate (Glu^0) in exchange for protonated GABA (GABA^+) at an external pH of 3.0 and lower (Ma et al., 2012, 2013; Tsai et al., 2013).

2. Lines 456 ff.:

“All cells of the *E. coli* population individually adapt to mild acid stress by activating the Gad system to varying degrees, and the glutamate decarboxylases GadA and GadB primarily might initially utilize intracellularly available glutamate under the consumption of protons. As described below, the antiporter GadC ~~is not active under mild acid stress~~ becomes more important under extreme acid stress. It should also be noted here that the decarboxylase GadB also undergoes a pH-dependent conformational change and exhibits an activity optimum at low pH (Capitani et al., 2003). Furthermore, glutamate is the most abundant intracellular metabolite, with a concentration of 100 mM, and its concentration is much higher compared to that of lysine (0.41 mM) and arginine (0.57 mM) (Bennett et al., 2009).

3. Lines 502-504: “The antiporter GadC preferentially transports protonated glutamate (Glu^- ~~or~~ Glu^+) in exchange for protonated GABA (GABA^+) only at an external pH of 3.0 and lower, ~~with no detectable activity at higher pH values~~ (Ma et al., 2012, 2013; Tsai et al., 2013).

5. For clarity, I'd suggest rephrasing lines 454-455 to "...cadaverine or agmatine respectively". Similarly, I'd rephrase lines 494-495 to read ",,only <0.01% of the population survives if the Gad system has not been pre-induced".

Thank you, both problems have been fixed (line 465 and line 506, respectively).

6. The "division of labour" model is interesting and fits the data well. The authors should however clarify how relevant this is in vivo where it is most likely to have evolved. In the gut, conditions will be very different to those described here - it will micro-aerobic or anaerobic, pH shock will be provided mostly by organic acids which have very different properties to inorganic ones (at least once cells have passed the stomach, which is far more acidic than the conditions described here), and *E. coli* cells will be a small minority of a complex microbial population, so the effects of secreted molecules may be minor. This is not to invalidate their model, but its limitations in understanding the behaviour of *E. coli* in its true habitat should be made clearer.

We agree with the thoughts of the reviewer and included the following new paragraph into the revised manuscript (lines 509 ff.).

"We can only speculate to what extent this model might hold up in a natural epitope of *E. coli*, such as digestive system of humans or animals. Before *E. coli* arrives in the intestine, it is exposed to an extreme acid stress with HCl in the stomach that this bacterium can only survive with the help of the Gad system. The mean pH in the human proximal small intestine is 6.6 and increases in the terminal ileum to 7.5. Then, there is a sharp fall in pH to 6.4 in the caecum, and an increase in colon with a final value of 7.0 (Evans et al., 1988). Depending on the availability of the amino acids lysine and arginine, the Cad and Adi systems might be switched on under these conditions. The latter system is increasingly activated due to the anaerobic conditions (Stim-Herndon et al., 1996). Both systems will still be heterogeneously distributed due to the low copy number of the main regulators CadC and AdiY. Fermentation products, such as acetate, also activate the Gad system (Zhao et al., 2018). The extent to which microcolonies of *E. coli* exist in the colon, in which common goods such as cadaverine and agmatine can change the micromilieu, is also unclear, as is the extent to which the host plays a role in acid stress. In fact, the levels of putrescine, a conversion product of agmatine, and cadaverine were found to be 15- and 3-fold, increased, respectively, in the inflamed mouse intestine mono-colonized with *E. coli* compared to an equally colonized healthy mouse cohort (Kitamoto et al., 2020)."

7. A minor point, but I don't find the data shown in Figure S1 panel B for *adiC* to be very convincing. There's no difference in cell counts between the 10⁻⁶ and 10⁻⁷ dilutions for *adiC::mCerulean*, and for a system where the effect is relatively subtle on pH sensitivity, I think plates with a larger number of colonies on should be used (ie a whole plate for each dilution, rather than a small spot). However, I don't think this alters the validity of the rest of the paper.

Thank you very much for raising this point. We performed the suggested experiment for the three-color strain *gadC:eGFP-adiC:mCerulean-cadB:mCherry* in LB pH 3.0 and pH 4.4, respectively. To investigate the functionality of *GadC:eGFP* and *AdiC:mCerulean*, the survival of cells was determined by counting the colony forming units after serial dilutions up to 3h. These new results are now presented in Figure S1b (see below) with a revised legend (lines 54-56 supplementary file). The experimental procedure is described in lines 736-739.

Reviewer #2 (Remarks to the Author):

The authors describe how *E. coli*'s three main acid-dependent amino-acid decarboxylation systems are induced under successive ranges of acidic pH. The triple fluorescent strain generates elegant data, although the essential phenomenon of these acid responses has been known for a while.

The authors do an excellent job of recounting known information in the Introduction and crediting appropriate sources. They are to be commended for not trying to claim they are the first to observed events reported back in the 90s.

What they do appear to show is the real-time kinetics of differential expression of the three systems. The heterogeneity of cell response is also shown. The authors might be interested to check Martinez et al 2012, which reports heterogeneity in cytoplasmic pH response to external acid shift: doi.org/10.1128/AEM.00354-12

Thank you for suggesting this publication, we included the results of this work in the introduction in lines 44-46 by adding the sentence "Despite this, pH homeostasis varies among individual bacterial cells as reported amongst others for *E. coli* and *Bacillus subtilis* (Martinez et al., 2012)."

One concern of this approach is that a cell with three fluorescent reporters may behave somewhat differently from the wild-type. However, the authors have some controls for these possibilities.

The functionality of the hybrid proteins was carefully checked and confirmed:
See Figure S1:

a) we constructed and tested different combinations of the three-color reporter strains: *gadC:eGFP-adiC:mCerulean-cadB:mCherry*, *gadC:mCherry-adiC:mCerulean:cadB-eGFP* and *gadC:mCerulean-adiC:mCherry-cadB:eGFP*
 b) we tested and confirmed the functionality of GadC:eGFP and AdiC:mCerulean in acid stress survival assays using LB medium at pH 3.0 and pH 4.4 (see also revised Figure S1).
 c) we confirmed the functionality of CadB:eGFP using a lysine-decarboxylase differential medium. Cells with an intact Cad-system increase the external pH due to the production and secretion of cadaverine (see Brameyer et al., 2020. PMID: 32482722)

Since the Gad system appears the most heterogeneous, the authors should comment on RpoS regulation which known to generate extensive genetic and transcriptional heterogeneity.

Good point: we inserted the following sentence in the discussion Lines 494-495:

“In addition, the stationary sigma factor RpoS is known to generate extensive transcriptional heterogeneity (Battesti et al., 2011; Heins et al., 2020; Hengge, 2011).”

For the figure legends: All the Y axes should be log scale; this would avoid the awkward breaks, and would be more appropriate for regulatory data, in which fold-change is generally the measure of on-off circuits.

We have changed Figure 6b accordingly to avoid breaks of the Y-axis (see below).

Reviewer #3 (Remarks to the Author):

Brameyer et al used a triple reporter strain to characterise cell individuality in the activation of three distinct responses to acid treatment in *E. coli*, namely glutamate decarboxylase, arginine decarboxylase, and lysine decarboxylase. They convincingly show that possessing all three systems confer an advantage to *E. coli* (and possibly several other species) that is therefore able to survive over a wide pH range. The manuscript is very well written and nicely presented with some beautiful fluorescence images (Figure 7 is superb!), the data are very robust.

Therefore, I believe that this manuscript should be published in Communications Biology after some minor adjustments as detailed below.

Line 35 please define AR

We defined AR in the abstract with “acid resistance” in line 35.

Lines 43-44: in the case of *E. coli* the intracellular pH does indeed change during standard growth in LB, please see <https://www.frontiersin.org/articles/10.3389/fmicb.2018.01739/full> (and references therein) and distinct *E. coli* subpopulations display different pH ranges possibly due to differential expression of the three AR systems reported in this manuscript, please see <https://journals.asm.org/doi/full/10.1128/mBio.00909-21> (and references therein)

We extended the introduction in lines 44-47 to highlight heterogenous pH homeostasis of *E. coli* cells and included the following sentences: “Despite this, pH homeostasis varies among individual bacterial cells as reported amongst others for *E. coli* and *Bacillus subtilis* (Martinez et al., 2012). Interestingly, persister cells of *E. coli* display a lower intracellular pH allowing survival after antibiotic treatment. (Goode et al., 2021).”

In our experimental set-up, cells were cultivated in buffered KE-medium (pH 7.6 or pH 5.8). Only to activate the Adi system, cells were cultivated in unbuffered LB medium at pH 4.4.

Lines 63-64: I found this sentence unclear because of the contrast between “mild acidic” and “extremely low”, perhaps rephrase?

Thank you for raising this point. We rephrased the sentence to “Activation of the Gad System occurs during the transition of an *E. coli* culture to stationary phase and during exponential growth in acidified media. Furthermore, this system is essential for cell survival at an extremely low pH of 2.5 (Bergholz et al., 2007; Biase et al., 1999; Castanie-Cornet et al., 1999; Weber et al., 2005).” (Lines 66-68).

Lines 126: could the authors please show data about the (lack of) spectra overlap between the three selected fluorophores

Thank you for raising this point. We now included the spectra of the three fluorophores (Figure S2d) as shown below and revised the legend (lines 64-70 and lines 90-91 supplementary file).

We had corrected the data for the overlap of the GFP (eGFP) and CFP (mCerulean) emission as described in Material and Methods, lines 694-696. The relative fluorescence of the GFP channel was corrected for the overlapping signal of the CFP channel via linear regression with the experimentally determined proportionality factor of 0.456, using the strain *gadC:mCerulean-cadB:eGFP* grown to stationary phase at pH 7.6. This condition induces GadC:mCerulean but not CadB:eGFP.

Line 135: this very nicely agrees with previous transcriptomic data <https://www.frontiersin.org/articles/10.3389/fmicb.2018.01739/full> (and references therein)

Indeed, this agrees quite nicely with the previous data. We therefore included these references in the introduction section by rephrasing the following sentence “Activation of the Gad system occurs during the transition of an *E. coli* culture to stationary phase and during exponential growth in acidified media. Furthermore, this system is essential for cell survival at an extremely low pH of 2.5 (Bergholz et al., 2007; Biase et al., 1999; Castanie-Cornet et al., 1999; Weber et al., 2005).” (Lines 66-68).

Lines 201-210 and Fig. 3: could the authors please add statistical significance to corroborate their Pearson’s correlation coefficients estimates? This is, for example, a built-in function in software such as prism graphpad.

Thank you for raising this point. The legend of Figure 3 now includes the p-values obtained by GraphPad Prism, which were used to calculate the Pearson’s correlation coefficients (lines 223, 227-229).

REVIEWERS' COMMENTS:

Reviewer #1 (Remarks to the Author):

All the points I raised in my initial review have been addressed.

Reviewer #2 (Remarks to the Author):

I approve the new manuscript with all the revisions provided.